# TROLL: TRUST REGIONS IMPROVE REINFORCEMENT LEARNING FOR LARGE LANGUAGE MODELS

**Philipp Becker**[1]*, **Niklas Freymuth**[1]*, **Serge Thilges**[1], **Fabian Otto**[2], **Gerhard Neumann**[1]
[1]Karlsruhe Institute of Technology,  [2]Microsoft Research

## ABSTRACT

Reinforcement Learning (RL) with PPO-like clip objectives has become the standard choice for reward-based fine-tuning of large language models (LLMs). Although recent work has explored improved estimators of advantages and normalization, the clipping mechanism itself has remained untouched. Originally introduced as a proxy for principled KL-based trust regions, clipping is a crude approximation that often causes unstable updates and suboptimal performance. We replace the clip objective with a novel discrete differentiable trust region projection, which provides principled token-level KL constraints. The projection operates on a sparse subset of the model's most important token logits to balance computational cost and projection effectiveness. Our approach, Trust Region Optimization for Large Language models (TROLL), serves as a direct replacement for PPO-like clipping during training and does not alter the model's inference behavior. Across mathematical reasoning and code generation tasks, model families, as well as advantage-estimation methods, TROLL consistently outperforms PPO-like clipping in terms of training speed, stability, and final success rates.

## 1 INTRODUCTION

Reinforcement Learning (RL) has become the standard approach for fine-tuning and aligning Large Language Models (LLMs) with preferences or verifiable rewards. For such post-training, the algorithms of choice are predominantly Proximal Policy Optimization (PPO)-style policy gradient approaches (Schulman et al., 2017). They first estimate an advantage function and then update the policy using an importance-weighted objective, clipped to prevent the ratio between new and old policies from deviating too much. Recent approaches such as GRPO (Shao et al., 2024), Dr.GRPO (Liu et al., 2025), GSPO (Zheng et al., 2025), and REINFORCE++ (Hu et al., 2025) improve the estimation of advantages and normalization, resulting in significant advances in RL for LLMs. Yet, all these approaches rely on PPO's clipping-based policy update mechanism.

Clipping is originally motivated via trust region methods (Schulman et al., 2015a; 2017), which provide a principled way to stabilize policy updates by constraining the KL divergence (Kullback & Leibler, 1951) between successive policies during training (Kakade & Langford, 2002; Peters et al., 2010). While theoretically sound, implementing such trust regions is often costly, in particular with modern LLMs whose vocabularies and thus output distributions can exceed $100\,000$ entries (Yang et al., 2025a;b). PPO instead clips the importance ratio to sidesteps this challenge. While empirically successful, it is a crude approximation of the underlying trust region (Wang et al., 2019; 2020). Crucially, it may lead to unstable optimization, poorly calibrated updates, as well as sensitivity to hyperparameters and implementation details, which often culminate in suboptimal performance (Engstrom et al., 2020; Andrychowicz et al., 2021; Otto et al., 2021; Huang et al., 2022).

As a remedy, we introduce Trust Region Optimization for Large Language models (*TROLL*) 🧌, a differentiable trust region projection approach that directly enforces token-level KL constraints between discrete distributions[1]. *TROLL* formulates a convex optimization problem that acts as a direct replacement to PPO-like clipping objectives. For each token, *TROLL* projects the output distribution of the new, updated policy onto a KL-trust region around the old policy that was used to sample the sequence. This process ensures that the new and old policies only differ by a given bound, preventing the policy update from diverging or collapsing. The left of Figure 1 shows a 3-dimensional example

---

*Equal contribution. Author order was decided by a fair coin flip.

[1]Project page, code, and datasets available at `https://niklasfreymuth.github.io/troll/`

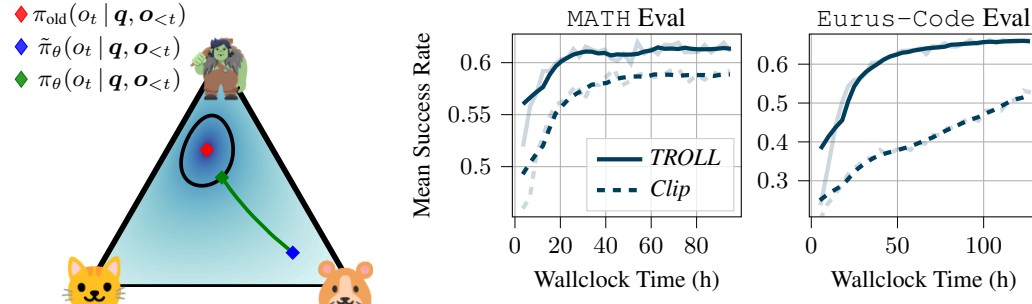

Figure 1: Trust Region Optimization for Large Language models (*TROLL*) overview. (**Left**) Example of a 3-token distribution (cat, troll, hamster). The old policy favors the troll, while the new policy shifts toward the hamster. The projection ensures that the updated policy stays within the trust region (circle). (**Right**) *TROLL* yields clear performance gains over PPO-like clipping (*CLIP*) on mathematical reasoning and code generation tasks, as shown for Qwen3-14B trained with GRPO.

where the old policy prefers the "troll" token, the new policy leans towards the "hamster" token, and the trust region constrains the update to keep the new policy close to the old one. The direction of the projection can be computed in closed form, while its step size is the solution to a one-dimensional convex Lagrangian dual problem. The projection leaves the new distribution unchanged if it already falls within the trust region, and can be efficiently solved in parallel in practice. *TROLL* enables differentiation through the solution of the projection problem using the OptNet framework (Amos & Kolter, 2017), which introduces only negligible computational overhead. This process maintains gradient information even for updates that are constrained by the trust region, in contrast to PPO-like clipping, which cuts gradients for tokens whose ratios exceed the clipping threshold. Further, the trust region is only effective during training and provides zero additional overhead during model inference. To incentivize the model to stay within the trust region for successive update steps, we additionally add a simple regression term between projected and unprojected tokens.

Applying *TROLL* directly to LLMs is computationally infeasible, since the large token vocabulary causes prohibitively expensive projections and memory overhead. However, natural language continuations and similarly LLM predictions are generally characterized by very few high-probability tokens (Zipf, 1949; Piantadosi, 2014; Kunstner et al., 2024; Duan et al., 2024; Ren & Sutherland, 2024). This property enables a sparsification scheme that discards unlikely tokens, retaining only the most relevant ones. On average, as few as $5-10$ tokens generally preserve more than $99.999\%$ of the distribution's probability mass. We thus modify *TROLL* to handle sparse distributions, allowing it to scale to modern LLMs and act as a drop-in replacement for PPO-style clipping.

We evaluate *TROLL* on Reinforcement Learning from Verifiable Rewards (RLVR) problems, focusing on mathematical reasoning. Using *TROLL* for GRPO (Shao et al., 2024) with models from the Qwen3 (Yang et al., 2025a) and Qwen2.5 (Yang et al., 2025b) families on DAPO-Math (Yu et al., 2025) yields substantial improvements in terms of training stability and success rate for evaluations based on the number of updates as well as wall clock time. Concretely, *TROLL* improves roughly $3-10$ percentage points, or $5-15\%$ relative over clipping for math-based reasoning. To assess robustness across algorithmic variants, we experiment with PPO (Schulman et al., 2017), Dr.GRPO (Liu et al., 2025), GSPO (Zheng et al., 2025), and REINFORCE++ (Hu et al., 2025). Across methods, *TROLL* consistently enables faster learning and improves success rates, indicating that its benefits are independent of the underlying advantage estimation method. We demonstrate similar improvements across additional math datasets, namely GSM8K (Cobbe et al., 2021a) and Eurus-2-RL-Math (Cui et al., 2025a), as well as models from the LLaMA 3 (Grattafiori et al., 2024), SmolLM3 (Bakouch et al., 2025), and Apertus (Hernández-Cano et al., 2025) families. Here, *TROLL* enables stable training for some models where while clipping fails entirely. Finally, we consider code generation using the Eurus-2-RL-Code (Cui et al., 2025a) dataset, showing improvements of $7-18$ percentage points, or $18-30\%$ relative over clipping across Qwen3 model sizes.

To summarize, we **i)** derive *TROLL*, a fully differentiable, principled trust region projection for discrete distributions that enforces per-token KL constraints, **ii)** introduce a sparsification scheme that lets the projection scale to LLM-scale vocabularies and implement it as a drop-in replacement for PPO-style heuristic clipping across RL algorithms, **iii)** demonstrate through experiments spanning different advantage-estimation methods, models and datasets that *TROLL* consistently improves success rates and training stability compared to clipping.

## 2 RELATED WORK

**Trust Regions in Reinforcement Learning.** Information-theoretic trust regions based on the KL divergence (Kullback & Leibler, 1951) are known to stabilize RL in classical (Kakade, 2001; Kakade & Langford, 2002; Peters et al., 2010; Abdolmaleki et al., 2015; Akrour et al., 2018) as well as modern deep learning settings (Schulman et al., 2015a; 2017). Trust Region Policy Optimization (TRPO) (Schulman et al., 2015a) limits the KL-divergence by solving a constrained optimization problem. PPO (Schulman et al., 2017) simplifies this approach to a first-order method, using a clipped surrogate objective to make policy optimization in RL scalable (Akkaya et al., 2019; Berner et al., 2019; Baker et al., 2020). However, PPO's trust region is less principled and very sensitive to practical implementation details (Engstrom et al., 2020; Andrychowicz et al., 2021; Huang et al., 2022). Several methods dynamically adapt PPO's clipping bounds (Wang et al., 2019; Xi et al., 2025), but still require clipping. Alternative methods (Abdolmaleki et al., 2018; Song et al., 2020) rely on regularizing the expected KL between subsequent policies. In contrast, *TROLL* enforces exact, token-wise trust regions through a differentiable KL projection, combined with adaptive strategies that control the trust region size.

**Trust Region Projections.** A different line of work (Otto et al., 2021; Akrour et al., 2019) enforces trust regions using projection-based methods. These approaches first compute the policy as usual, and then project it back into a feasible set defined by a trust region constraint. In particular, Otto et al. (2021) compute exact trust region projections for each state when using Gaussian policies, leading to improvements in high-dimensional action spaces (Celik et al., 2024; Li et al., 2024a; Hoang et al., 2025; Otto et al., 2025). Similar to us, they rely on Lagrangian optimization (Boyd & Vandenberghe, 2004) and implicit differentiation (Amos & Kolter, 2017), but focus on Gaussian distributions and continuous control tasks (Brockman et al., 2016). *TROLL* instead proposes differentiable projections for categorical distributions and provides an efficient implementation via a sparsification scheme. This combination lets *TROLL* scale to modern-day LLMs while preserving the stability of classical trust region methods.

**Reinforcement Learning for LLMs.** Recently, RL has become a key tool in the post-training stage of LLMs. Popular frameworks include RL from human feedback (RLHF) (Christiano et al., 2017; Ziegler et al., 2019; Stiennon et al., 2020; Ouyang et al., 2022) for LLM alignment and RLVR (Luong et al., 2024; Lambert et al., 2024) for reasoning tasks such as mathematical problem solving or code generation. In RLHF, preference-based methods often optimize a reward while remaining close to a reference policy via an added expected KL penalty. This reference policy is typically the supervised fine-tuning model (Stiennon et al., 2020; Ouyang et al., 2022). For example, Direct Preference Optimization (DPO) (Rafailov et al., 2023) optimizes this objective in closed form on preference data, thereby avoiding policy rollouts. *TROLL* does not constrain the updates to a fixed reference, instead enforcing proximity to the previous step's policy to stabilize on-policy optimization. While this work focuses on RLVR for LLMs, *TROLL* can also be applied to other LLM post-training settings, and more generally, discrete RL tasks.

Several recent methods avoid PPO's importance sampling and thus clipping entirely. These methods combine group rewards with a standard policy gradient objective (Chu et al., 2025), utilize an additional value network (Richemond et al., 2024), use Q-learning (Clavier et al., 2025), or employ a contrastive loss to enable off-policy learning (Flet-Berliac et al., 2024; Cohere et al., 2025). However, while these approaches are promising, current research on RLVR still often relies on advantage estimation. While not the focus of this work, we briefly show that one such non-clipping-based approach also benefits from *TROLL*, since its trust regions directly act on the LLM's policy.

**Advantage Estimation for LLM Post-Training.** PPO (Schulman et al., 2017) requires an expensive explicit value model. Especially for RLVR, where evaluating multiple rollouts per input is comparatively cheap, sample-based advantage estimation methods like Group-Relative Policy Optimization (GRPO) (Shao et al., 2024) have become popular alternatives. Variants include GRPO Done Right (Dr.GRPO) (Liu et al., 2025), which addresses GRPO's optimization biases that favor longer responses, and REINFORCE++ (Hu et al., 2025), which uses Global Advantage Normalization (Andrychowicz et al., 2021) to improve training stability. Group Sequence Policy Optimization (GSPO) (Zheng et al., 2025) extends importance ratios from token to sequence level to improve update stability. However, all these methods still depend on PPO-style clipping to stabilize policy updates. We introduce *TROLL* as a more principled drop-in replacement, applicable regardless of how advantages are computed and compatible with all the approaches above.

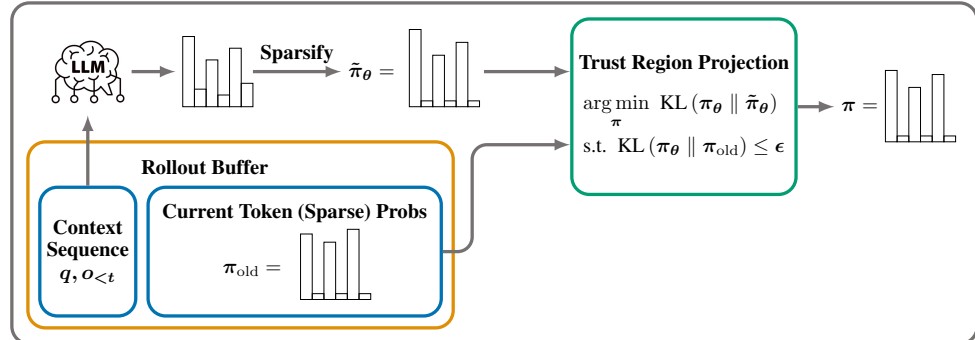

Figure 2: *TROLL* replaces PPO-like clipping with a differentiable trust region projection approach. Given the current output distribution of an LLM $\tilde{\pi}_\theta(o_t \,|\, \boldsymbol{q}, \boldsymbol{o}_{<t})$, and the distribution that was used to collect the sequence $\pi_{\text{old}}(o_t \,|\, \boldsymbol{q}, \boldsymbol{o}_{<t})$ for the replay buffer, *TROLL* enforces a per-token trust region by solving Equation 3 for each token. The resulting distribution is then used in the RL objective (Equation 5) to update the LLM parameters. To scale this approach to the vocabulary size of modern LLMs, *TROLL* uses a sparsification approach, which allows working on a small subset of logits while retaining most of the distribution's mass.

## 3 TRUST REGION OPTIMIZATION FOR LARGE LANGUAGE MODELS

Reinforcement Learning (RL) for Large Language Models (LLMs) finetunes the LLM's parameters, $\theta$, using policy ratio objectives (Schulman et al., 2015a) of the form

$$\mathcal{J}_{\text{ratio}}(\theta) = \mathbb{E}_{\boldsymbol{o}\sim\pi_{\text{old}}(\boldsymbol{o}|\boldsymbol{q})\mathcal{D}(\boldsymbol{q})} \left[ \frac{1}{|\boldsymbol{o}|} \sum_{t=1}^{|\boldsymbol{o}|} \left( \frac{\tilde{\pi}_\theta(o_t \,|\, \boldsymbol{q}, \boldsymbol{o}_{<t})}{\pi_{\text{old}}(o_t \,|\, \boldsymbol{q}, \boldsymbol{o}_{<t})} A(o_t, \boldsymbol{q}, \boldsymbol{o}_{<t}) \right) \right], \tag{1}$$

where $\tilde{\pi}_\theta(o_t \,|\, \boldsymbol{q}, \boldsymbol{o}_{<t})$ is the probability of the sampled token under the current LLM policy. Here, $\pi_{\text{old}}(o_t \,|\, \boldsymbol{q}, \boldsymbol{o}_{<t})$ is the token's probability under the LLM policy that was used for data collection in the previous iteration. The context sequence consists of the prompt $\boldsymbol{q}$ and prior response tokens $\boldsymbol{o}_{<t}$. The advantage estimate $A_t = A(o_t, \boldsymbol{q}, \boldsymbol{o}_{<t})$ measures if a token is better or worse than the average behavior. Thus, maximizing $\mathcal{J}_{\text{ratio}}(\theta)$ increases the probability of good responses while decreasing the probability of bad ones. In practice, we estimate $A_t$ using an explicit value model as in PPO (Schulman et al., 2017) or purely sample-based as in GRPO and its variants (Shao et al., 2024; Liu et al., 2025; Zheng et al., 2025). For such policy ratio objectives, stable and effective optimization requires keeping $\tilde{\pi}_\theta(o_t \,|\, \boldsymbol{q}, \boldsymbol{o}_{<t})$ and $\pi_{\text{old}}(o_t \,|\, \boldsymbol{q}, \boldsymbol{o}_{<t})$ close, so that the importance ratio $r_{\theta,t} = \tilde{\pi}_\theta(o_t \,|\, \boldsymbol{q}, \boldsymbol{o}_{<t})/\pi_{\text{old}}(o_t \,|\, \boldsymbol{q}, \boldsymbol{o}_{<t})$ remains close to one (Schulman et al., 2015a; 2017). PPO attempts to maintain this proximity by clipping the ratio,

$$\mathcal{J}_{\text{ppo}}(\theta) = \mathbb{E}_{o_t\sim\pi_{\text{old}}(\boldsymbol{o}|\boldsymbol{q})\mathcal{D}(\boldsymbol{q})} \left[ \frac{1}{|\boldsymbol{o}|} \sum_{t=1}^{|\boldsymbol{o}|} \min\left( r_t A_t; \text{clip}\left(r_t, 1 - \epsilon_{\text{ppo}}, 1 + \epsilon_{\text{ppo}}\right) A_t \right) \right]. \tag{2}$$

However, this clipping is a crude approximation of the underlying trust region principle. While it prevents large updates, this approach is purely heuristic and suppresses gradients when the ratio falls outside the clipping range, resulting in unstable and inefficient learning. In contrast, token-wise KL-based constraints offer a principled approach to limit the change between successive policies. Our method, Trust Region Optimization for Large Language models (*TROLL*), implements these constraints using differentiable trust region projections (Otto et al., 2021) as a drop-in replacement for the PPO-like clipping.

### 3.1 DISCRETE DIFFERENTIABLE TRUST REGION PROJECTIONS

**Trust Region Projection.** Our projection formally solves the convex optimization problem

$$\pi_\theta(o_t \,|\, \boldsymbol{q}, \boldsymbol{o}_{<t}) = \underset{\hat{\pi}_\theta(o_t \,|\, \boldsymbol{q}, \boldsymbol{o}_{<t})}{\arg\min} \ \text{KL}\left( \hat{\pi}_\theta(o_t \,|\, \boldsymbol{q}, \boldsymbol{o}_{<t}) \,\|\, \tilde{\pi}_\theta(o_t \,|\, \boldsymbol{q}, \boldsymbol{o}_{<t}) \right) \tag{3}$$

$$\text{s.t. KL}\left( \hat{\pi}_\theta(o_t \,|\, \boldsymbol{q}, \boldsymbol{o}_{<t}) \,\|\, \pi_{\text{old}}(o_t \,|\, \boldsymbol{q}, \boldsymbol{o}_{<t}) \right) \leq \epsilon$$

for every output token $o_t$[2]. Intuitively, the projection finds the policy distribution closest to the current LLM policy $\tilde{\pi}_\theta(o_t \mid \boldsymbol{q}, \boldsymbol{o}_{<t})$ while remaining within an $\epsilon$-bound of the policy used to collect the data for the current iteration $\pi_{\text{old}}(o_t \mid \boldsymbol{q}, \boldsymbol{o}_{<t})$. The solution to this optimization problem is derived in Appendix A.1 and given as

$$\pi_\theta(o_t \mid \boldsymbol{q}, \boldsymbol{o}_{<t}) \propto \exp\left(\frac{\eta^* \log \pi_{\text{old}}(o_t \mid \boldsymbol{q}, \boldsymbol{o}_{<t}) + \log \tilde{\pi}_\theta(o_t \mid \boldsymbol{q}, \boldsymbol{o}_{<t})}{\eta^* + 1}\right), \tag{4}$$

which is a geometric interpolation between the logits of $\tilde{\pi}_\theta(o_t \mid \boldsymbol{q}, \boldsymbol{o}_{<t})$ and $\pi_{\text{old}}(o_t \mid \boldsymbol{q}, \boldsymbol{o}_{<t})$. Here, $\eta^*$ acts as a step size controlling how far the projection moves the new policy to the old one. For each token, we can compute the optimal $\eta^*$ which enforces the trust region constraint by solving the convex dual of Equation 3. This dual is a scalar optimization problem, which we derive and state in Appendix A.2, and can be solved with sufficient accuracy using a few iterations of ternary, or more generally, $n$-ary, bracketing. Furthermore, projecting is only necessary if the trust region bound is violated, which is only the case for very few, but highly relevant tokens. Thus, we can avoid it for the vast majority of tokens by filtering them beforehand.

**Policy Updates with Trust Region Projections.** After projection, the policy $\pi_\theta(o_t \mid \boldsymbol{q}, \boldsymbol{o}_{<t})$ satisfies the trust region constraint and can be used to optimize the LLM parameters $\theta$, via an update similar to Equation 1. However, the LLM output $\tilde{\pi}_\theta(o_t \mid \boldsymbol{q}, \boldsymbol{o}_{<t})$ may still deviate arbitrarily from the old policy, complicating inference and successive updates. To avoid this, we follow Otto et al. (2021) and address this by regressing the LLM output $\tilde{\pi}_\theta(o_t \mid \boldsymbol{q}, \boldsymbol{o}_{<t})$ toward its projection $\pi_\theta(o_t \mid \boldsymbol{q}, \boldsymbol{o}_{<t})$, resulting in an objective

$$\mathcal{J}_{\text{Troll}}(\theta) = \tag{5}$$

$$\mathbb{E}_{o_t \sim \pi_{\text{old}}(\boldsymbol{o} \mid \boldsymbol{q}) \mathcal{D}(\boldsymbol{q})} \left[ \frac{1}{|\boldsymbol{o}|} \sum_{t=1}^{|\boldsymbol{o}|} \left( \frac{\pi_\theta(o_t \mid \boldsymbol{q}, \boldsymbol{o}_{<t})}{\pi_{\text{old}}(o_t \mid \boldsymbol{q}, \boldsymbol{o}_{<t})} A_t \right) - \alpha \text{KL}\left( \tilde{\pi}_\theta(o_t \mid \boldsymbol{q}, \boldsymbol{o}_{<t}) \,\|\, \lfloor \pi_\theta(o_t \mid \boldsymbol{q}, \boldsymbol{o}_{<t}) \rfloor \right) \right],$$

where $\lfloor \ \rfloor$ denotes the stop gradient operator and $\alpha$ is a user-specified regression weight. Crucially, the projected policy $\pi_\theta(o_t \mid \boldsymbol{q}, \boldsymbol{o}_{<t})$ is used to compute the ratios and as a regression target for the LLM output $\tilde{\pi}_\theta(o_t \mid \boldsymbol{q}, \boldsymbol{o}_{<t})$. For the regression, we stop the gradients through $\pi_\theta(o_t \mid \boldsymbol{q}, \boldsymbol{o}_{<t})$ so that the LLM policy $\tilde{\pi}_\theta(o_t \mid \boldsymbol{q}, \boldsymbol{o}_{<t})$ is pulled towards the output of the projection $\pi_\theta(o_t \mid \boldsymbol{q}, \boldsymbol{o}_{<t})$, not the other way around. The regression term only affects projected tokens and still allows policy updates up to the KL bound, making the approach robust to the choice of $\alpha$. We thus set to $\alpha = 1$ in all experiments for simplicity. Notably, our objective in Equation 5 makes no assumption on the advantages $A_t$. Thus, *TROLL* can be directly applied to a variety of existing advantage estimation methods, including PPO, GRPO, Dr.GRPO, and GSPO. Algorithm 1 provides pseudocode.

**Making Trust Region Projections Differentiable.** To propagate gradients through our projection, we can rely on autograd tools such as PyTorch (Paszke et al., 2019), except for the numerical optimization of the dual. Formally, the optimal $\eta^*$ is a function of the LLM policy $\tilde{\pi}_\theta(o_t \mid \boldsymbol{q}, \boldsymbol{o}_{<t})$. To obtain a fully differentiable projection, we need the gradient $\partial \eta^* / \partial \tilde{\pi}_\theta(o_t \mid \boldsymbol{q}, \boldsymbol{o}_{<t})$, which describes how the LLM output influences the optimal step size. We follow the OptNet framework (Amos & Kolter, 2017) and differentiate the KKT conditions (Karush, 1939; Boyd & Vandenberghe, 2004) of the dual solution via implicit differentiation (Dontchev & Rockafellar, 2009) and differential calculus (Magnus & Neudecker, 1989). This approach lets us compute a closed-form gradient instead of differentiating through the numerical optimization. Appendix A.3 provides derivations. Appendix B provides code snippets and a schematic of our projection's compute graph, including gradients.

## 3.2 SPARSE AND EFFICIENT REPRESENTATIONS OF TOKEN DISTRIBUTIONS

Naively implementing *TROLL* requires storing and projecting the full vocabulary distribution for each token. Using Qwen3's tokenizer (Yang et al., 2025b) as an example, this results in an overhead of 151 936 logits per token, which is prohibitively expensive. To address this issue, we sparsify both the distributions and the implementation of the projection. We greedily select the $K$ tokens with the largest probability mass, sort them by their mass, and then only retain those needed to cover a cumulative mass of $1 - \delta$. We additionally always keep the token actually selected by the LLM policy

---

[2]$\pi_\theta(o_t \mid \boldsymbol{q}, \boldsymbol{o}_{<t})$ must also remain a valid distribution, i.e., $\sum_{o_t} \pi_\theta(o_t \mid \boldsymbol{q}, \boldsymbol{o}_{<t}) = 1$ and $\pi_\theta(o_t \mid \boldsymbol{q}, \boldsymbol{o}_{<t}) \geq 0$ for all $o_t$. We omit these constraints for brevity and elaborate in Appendix A.

---

**Algorithm 1** Optimizing LLMs with TROLL

---

1: LLM policy $\tilde{\pi}_\theta(o_t \mid \boldsymbol{q}, \boldsymbol{o}_{<t})$, Training data $\mathcal{D}$, KL Bound $\epsilon$
2: **for** step $s=1 \ldots$ **do**
3:     Sample batch of questions $\boldsymbol{q} \sim \mathcal{D}$
4:     Sample responses $\{\boldsymbol{o} \sim \tilde{\pi}_\theta(o_t \mid \boldsymbol{q}, \boldsymbol{o}_{<t})$ using the LLM policy.
5:     Set reference policy $\pi_{\text{old}} = \tilde{\pi}_\theta$
6:     Sparsify and save corresponding logits $\pi_{\text{old}}(o_t \mid \boldsymbol{q}, \boldsymbol{o}_{<t})$
7:     **for** minibatch $b$ **do**
8:         Compute and sparsify current logits $\tilde{\pi}_\theta(o_t \mid \boldsymbol{q}, \boldsymbol{o}_{<t})$
9:         Estimate advantages $A_t$ using any advantage estimation method
10:         **for** response tokens $o$ in parallel **do**
11:             **if** KL $(\tilde{\pi}_\theta(o_t \mid \boldsymbol{q}, \boldsymbol{o}_{<t}) \parallel \pi_{\text{old}}(o_t \mid \boldsymbol{q}, \boldsymbol{o}_{<t})) \leq \epsilon$ **then**
12:                 Set $\pi_\theta(o_t \mid \boldsymbol{q}, \boldsymbol{o}_{<t}) = \tilde{\pi}_\theta(o_t \mid \boldsymbol{q}, \boldsymbol{o}_{<t})$         ▷ no projection needed
13:             **else**
14:                 Compute $\eta^*$ by numerically optimizing Equation 13.
15:                 Compute $\pi_\theta(o_t \mid \boldsymbol{q}, \boldsymbol{o}_{<t})$ using Equation 4     ▷ project to $\pi_{\text{old}}(o_t \mid \boldsymbol{q}, \boldsymbol{o}_{<t})$
16:             **end if**
17:         **end for**
18:         Update LLM parameters $\theta$ using Equation 5
19:     **end for**
20: **end for**

---

to ensure gradient information for this token. The top-$K$ filtering both upper bounds the number of kept logits, acting as a fail-safe to prevent excessive memory usage for high-entropy predictions, and allows for efficient sorting of relevant tokens. Since pre-trained LLMs generally have very low perplexity (Kaplan et al., 2020; Hoffmann et al., 2022; Ruan et al., 2024), this thresholding allows us to maintain almost all of the probability mass of the logit distribution with very few average kept logits. Empirically, using $K=64$ and $\delta=10^{-5}$ usually allows us to keep $99.999\%$ of probability mass with only $5-10$ average tokens for most of the tested model and task combinations. Finally, for the discarded tokens, we cannot assume a probability of truly 0 but have to use a small default mass $p_d > 0$ to maintain well-behaved distributions. After sparsification, we re-normalize the kept tokens with Equation 21, taking into account the number of non-kept tokens and default mass. We perform the sparsification in chunks of the full generated sequences to prevent memory spikes.

While greedily keeping the highest-probability tokens is intuitively useful, we additionally show in Theorem A.1 in Appendix A.4 that it yields best possible KL approximation under mild assumptions. Additionally, under moderate assumptions, the error introduced by sparsification is bound by

$$\text{KL}\left(p \parallel q\right) \leq \gamma^{-1} \text{KL}\left(p' \parallel q'\right) + \delta \log \frac{\delta}{q_{\min}}, \tag{6}$$

where $p$ and $q$ are arbitrary categorical distributions, $p', q'$ the corresponding sparsified distributions, $q_{\min} \leq q(x_i)$ denotes a reference lower bound and $\gamma \approx 1$ the renormalization constant. Theorem A.2 provides the proof and demonstrates that, for the hyperparameters used in Qwen3, the error incurred by enforcing the trust region on the sparsified distributions rather than on the full distributions is approximately two orders of magnitude smaller than the bound itself. The sparsification reduces memory and computation cost to the point where *TROLL* only incurs minimal overhead on modern LLMs, making it a practically viable alternative to PPO-like clipping. Further, this overhead is constant in model size, causing its relative cost to diminish for larger models. Figure 2 shows a schematic overview of *TROLL*'s training, and Section 6 provides additional analysis of the sparsification and projection behavior in practice.

## 4 EXPERIMENTS

**Datasets.** We evaluate *TROLL* by finetuning LLMs for mathematical reasoning and code generation using an Reinforcement Learning from Verifiable Rewards (RLVR) setup. DAPO-Math (Yu et al., 2025) consists of 17 thousand math questions and answers that are obtained from web scraping and manual annotation. Appendix F provides an example question. We randomly split off 1024 samples to provide an in-distribution evaluation dataset, and use the remaining samples for training. We refer to those sets as `DAPO`-Eval and `DAPO`-Train, respectively. Additionally, we follow the evaluation

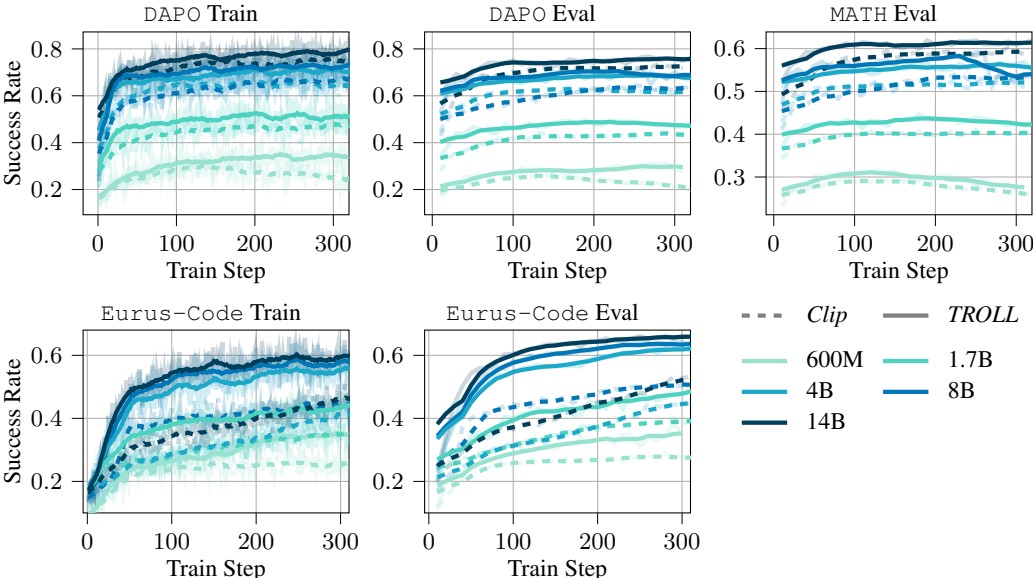

Figure 3: Comparison of *TROLL* (full lines) and *Clip* (dashed lines) across GRPO-trained Qwen3 models with 600M to 14B parameters for `DAPO` (**top**) and `Eurus-Code` (**bottom**). Full-opacity lines mark smoothed results, while the background shows original values. *TROLL* consistently boosts training efficiency and final success rates on math-related questions and code generation tasks. These gains translate to different evaluation datasets.

setup of Cui et al. (2025b) and use a suite of test datasets, which we call `Math`-Eval, comprised of MATH500 (Hendrycks et al., 2021), AMC, AIME2024 (Li et al., 2024b), AIME 2025, OMNI-MATH (Gao et al., 2025), OlympiadBench (He et al., 2024), and Minerva (Lewkowycz et al., 2022). As in previous work (Cui et al., 2025b), we report the mean of 32 rollouts for the comparatively small AIME2024, AIME2025, and AMC datasets to reduce evaluation variance. `GSM8K` (Cobbe et al., 2021b) contains grade school math problems with final integer answers, consisting of 8.5k training and 1.3k test problems. Finally, we consider the `Eurus-2-RL` data (Cui et al., 2025a), which contains mathematical reasoning and code generation tasks. We consider those separately, resulting in `Eurus-Math` and `Eurus-Code`. For both datasets, we use the train and validation sets as is.

The three mathematical reasoning tasks range from comparatively simple grade school problems to complex math olympiad tasks, and the code generation demands understanding of techniques such as dynamic programming, data structures and amortized analysis. Appendix D.3 provides further details on all datasets and the reward calculations.

**Models.** We experiment with Qwen3-{0.6B, 1.7B, 4B, 8B, 14B} (Yang et al., 2025a), which we use in thinking mode, and Qwen2.5-{0.5B,1.5B,3B,7B}-Instruct (Yang et al., 2025b). Furthermore, we include both the instruct and non-instruct versions of Llama-3.1-8B, Llama-3.2-3B (Grattafiori et al., 2024), and Apertus-8B (Hernández-Cano et al., 2025). Finally, we include Smol-LM3-3B Bakouch et al. (2025) and a version of Llama fine-tuned on FineMath (HuggingFaceTB, 2025). These models range from 500M to 14B parameters and cover different vocabulary sizes, tokenizers, model architectures, pre-training paradigms, and datasets, as well as initial math capabilities.

**Methods.** We focus on GRPO (Shao et al., 2024) due to its popularity and empirical success. We also include PPO (Schulman et al., 2017), which uses an explicit value model for Generalized Advantage Estimation (Schulman et al., 2015b), and three additional GRPO variants, namely Dr.GRPO (Liu et al., 2025), GSPO (Zheng et al., 2025), and REINFORCE++ (RF++) (Hu et al., 2025). All methods differ in their advantage estimation and in their normalization of Equation 1, yet they all rely on PPO-like clipping, which makes them amenable to using *TROLL*. We compare each method's original clipping-based version to using *TROLL* projections, denoted as (*Clip*) and (*TROLL*), respectively. Finally, we compare to BAPO (Xi et al., 2025) as an adaptive clipping method, and GPG (Chu et al., 2025) as a clipping-free baseline. For the latter, we compare to the vanilla variant (GPG) and also add the *TROLL* projection to its policy update (GPG (*TROLL*)).

|  |  | Qwen3-8B | | | | | Qwen2.5-7B-Instruct | | | | |
|---|---|---|---|---|---|---|---|---|---|---|---|
|  |  | GRPO | Dr.GRPO | PPO | GSPO | RF++ | GRPO | Dr.GRPO | PPO | GSPO | RF++ |
| DAPO | *Clip* | 0.667 | 0.678 | 0.640 | 0.000 | 0.648 | 0.443 | 0.467 | 0.444 | 0.159 | 0.429 |
| Train | *TROLL* | 0.721 | 0.704 | 0.744 | 0.736 | 0.742 | 0.495 | 0.513 | 0.431 | 0.481 | 0.486 |
| DAPO | *Clip* | 0.640 | 0.653 | 0.602 | 0.000 | 0.626 | 0.323 | 0.331 | 0.324 | 0.093 | 0.323 |
| Eval. | *TROLL* | 0.691 | 0.674 | 0.715 | 0.706 | 0.728 | 0.389 | 0.389 | 0.353 | 0.390 | 0.380 |
| MATH | *Clip* | 0.541 | 0.549 | 0.508 | 0.000 | 0.520 | 0.313 | 0.317 | 0.319 | 0.127 | 0.311 |
| Eval. | *TROLL* | 0.551 | 0.546 | 0.591 | 0.580 | 0.578 | 0.350 | 0.359 | 0.349 | 0.333 | 0.344 |

Table 1: Final train and evaluation success rates on `DAPO` for Qwen3-8B and Qwen2.5-7B-Instruct methods for different advantage estimation methods for *TROLL* and *Clip*. The better approach is marked in blue. *TROLL* significantly improves over *Clip* in most cases, and is able to successfully train GSPO, where *Clip* causes divergence and little to no success rates on both models.

**Experiment Setup.** We base our experiments on the `verl` repository[3], using default parameters and training recipes where applicable. We set the group size for the advantage normalization of all methods to 8. We use a token-level loss aggregation (Yu et al., 2025) for PPO and GRPO, and opt for method-specific loss aggregations for Dr.GRPO and GSPO. We evaluate the test datasets every 10 steps. To improve visibility, we use sliding windows of size 7 and 21 for the train and test evaluations, respectively, while showing the unsmoothed values in the background. Appendix C provides additional details, including hyperparameters in Table 3. Appendix D shows all results.

## 5 RESULTS

**Qwen Experiments on `DAPO-Math`.** We evaluate models from the Qwen3 and Qwen2.5-Instruct families (Yang et al., 2025b;a) on `DAPO` (Yu et al., 2025). The top of Figure 3 compares *TROLL* and the *Clip* objective for different Qwen3 model sizes optimized with GRPO (Shao et al., 2024). *TROLL* consistently improves training performance, causing more sample-efficient training and improved success rates at convergence for all models. These results directly translate to both evaluation sets, `DAPO`-Eval and `MATH`-Eval. Interestingly, the 4B *TROLL* model almost matches the performance of the 14B *Clip* one. Figure 7 in Appendix D.1 shows similar performance trends across Qwen2.5-Instruct model sizes. The right of Figure 1 further compares the runtime of both variants on Qwen3-14B, showing that *TROLL*'s projections do not incur a significant computational overhead. Finally, Appendix F provides example sequences generated by Qwen3-14B on a `MATH`-Eval problem.

Table 1 compares *TROLL* and *Clip* results for Qwen3-8B and Qwen2.5-7B-Instruct for GRPO, Dr.GRPO, PPO, GSPO, and RF++. *TROLL* generally improves success rates by 3-10 percentage points across methods and evaluated datasets. In these experiments, choosing *TROLL* over *Clip* is usually more beneficial than selecting any of the considered advantage estimation methods. Table 4 provides results for the individual `MATH` datasets, while Figure 8 and Figure 9 show full training curves for Qwen3-8B and Qwen2.5-7B-Instruct, respectively. Both figures show that GSPO (*Clip*) eventually diverges during training, while GSPO (*TROLL*) remains stable and achieves similar success rates to the other advantage estimation methods.

**Qwen Experiments on `Eurus-Code`.** The bottom of Figure 3 shows that *TROLL* is directly applicable to code generation tasks, yielding substantial advantages over *Clip* for all model sizes. For instance, our Qwen3-1.7B (*TROLL*) improves over Qwen3-4B (*Clip*), and significantly outperforms the model of its own size. Specifically, we see improvements of $7-18$ percentage points, which translate to a $18-30\%$ relative gain, when using *TROLL* instead of *Clip*.

**Additional Model Families.** Figure 4 shows various models of different families and sizes on `GSM8K`, again indicating a clear benefit for *TROLL* over the *Clip* objective. Here, models of the Llama family often need a significant number of training steps before *Clip* shows a positive training signal, while *TROLL* causes the models to start learning much faster. *TROLL* further enables stable and fast training for Apertus, while *Clip* fails in some setups. Appendix D.2 provides additional results on more models and the `GSM8K` dataset. We omit evaluations for `DAPO` with models other than SmolLM3-3B, as none matched the performance of Qwen3-1.7B in preliminary *Clip* experiments.

---

[3] https://github.com/volcengine/verl

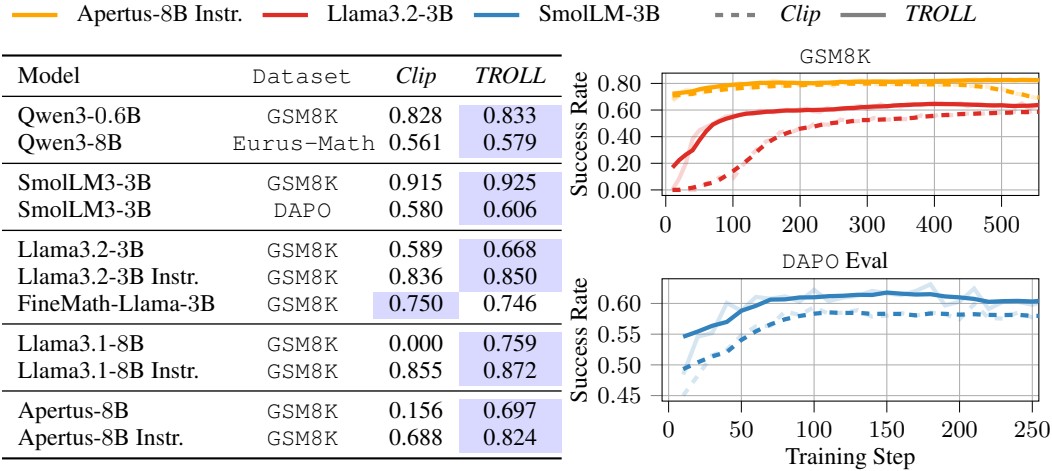

Figure 4: (**Left**) Final evaluations for *TROLL* and *Clip* for different combinations of models and datasets trained with GRPO. The better approach between *TROLL* and *Clip* is marked in blue. (**Right**) Comparison of *TROLL* (full lines) and the *Clip* objective (dashed lines) for different models trained with GRPO. *TROLL* generally improves over *Clip*, and performs well across all considered datasets. In particular, *TROLL* leads to significantly faster learning for different Llama models, where *Clip* often takes significantly more iterations to obtain a positive training signal. *TROLL* also showcases more stable performance compared to *Clip* throughout training.

**Additional Math Datasets.** Considering other math datasets, the top rows on the left of Figure 4 shows that *TROLL* is also beneficial on other datasets, as evaluated on `Eurus-Math` for Qwen3-8B and the simpler `GSM8K` for Qwen3-0.6B. Appendix D.3 provides detailed success rates for `Eurus-Math` in Figure 13 and additional results on `GSM8K` for larger Qwen3 models in Figure 12.

**Additional Training Algorithms.** Figure 14 in Appendix D.4 shows that adaptive clipping via BAPO (Xi et al., 2025) slightly improves over regular *Clip* on evaluation data, but still clearly underperforms *TROLL*. It also shows preliminary results on how *TROLL* can benefit non-clipping-based policy gradient methods such as GPG (Chu et al., 2025). While vanilla GPG suffers from stability issues in our experiments, these issues are resolved by adding our differentiable trust region projections, resulting in performance comparable to GRPO (*TROLL).*

## 6 ANALYSIS AND ABLATIONS

**KL Bounds and Sparsity Threshold Experiments.** We explore different values for the KL bound $\epsilon$ and the maximum number of kept tokens $K$ in the sparsification process for Qwen3-8B trained with GRPO on the `DAPO` dataset. The left of Figure 5 finds that a lower KL slows down training but does not affect convergence, while a higher KL leads to worse success rates, likely due to too large policy updates. A small $K=16$ causes poor updates, presumably due to poor estimates of the underlying dense distributions, while a larger $K=256$ increases cost but does not improve over our default $K=64$. Figure 15 in Appendix E.1 provides additional detail. These results suggest that *TROLL* requires an accurate KL projection, while showing that there is a wide range of suitable hyperparameters for both the KL bound and the sparsification. Finally the top row of Figure 18 shows that $5-10$ tokens are usually sufficient to capture $99.999\%$ of logit probabilities.

**Batch Size.** Figure 16 in Appendix E.2 compares the training behavior of *TROLL* and *Clip* for different batch sizes. We find that *TROLL* can easily deal with larger batch sizes and thus less recent data in its optimization, while *Clip* clearly degrades when increasing the batch size.

**Output Diversity and Entropy.** Recent work has shown that the PPO-like *Clip* objective tends to exploit the LLM's existing knowledge by reducing each token distribution's entropy to increase the reward (Cui et al., 2025b). In contrast, the bottom right of Figure 5 shows that *TROLL* preserves entropy. Additionally, Figure 17 in Appendix E.3 shows a clear correlation between *TROLL*'s ability to preserve entropy and improve performance on `Eurus-Code`.

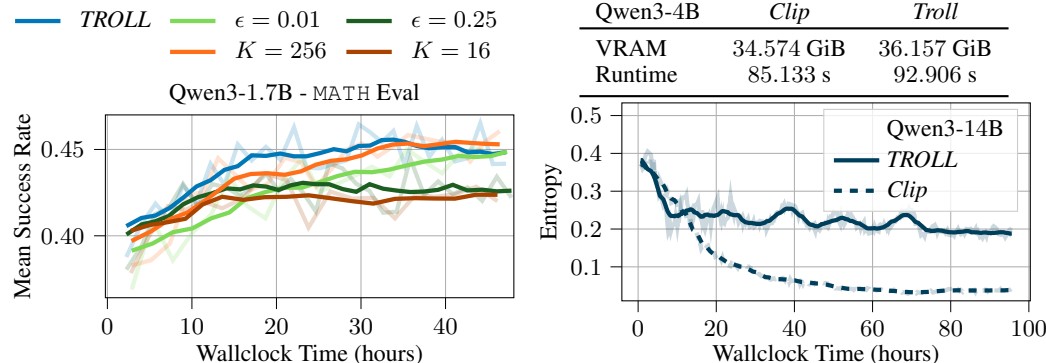

Figure 5: (**Left**) Qwen3-1.7B trained with GRPO using the *TROLL* projection compared to different hyperparameter choices. *TROLL* works well for conservative KL bounds $\epsilon$ and top-$K$ logit selections, but is slower for too conservative values and degrades slightly for too aggressive updates or token pruning. (**Top Right**) Memory and runtime comparison between *TROLL* and *Clip* in a controlled environment. *TROLL* imposes a modest overhead compared to the cost of training the LLM parameters. (**Bottom Right**) *TROLL* generally maintains more entropy during training while showing higher success rates when compared to *Clip*, as shown for Qwen3-14B.

**Projection Fraction.** Comparing the fraction of clipped tokens for *Clip* with the fraction of projected tokens for *TROLL* shows that both trust region approaches roughly affect the same number of tokens. The observed stability improvements are thus not merely caused by more restrained tokens. We compare both ratios for larger Qwen3 models in the middle row of Figure 18.

**Response Length.** The lower row of Figure 18 shows that *TROLL* adapts response length more quickly to ranges suitable for solving the tasks. This faster adaption reflects the faster performance improvements achieved by *TROLL*.

**Computational Overhead.** Appendix E.4 provides a controlled experiment setup for measuring *TROLL*'s computational overhead. We find on the top right table of Figure 5 that the memory overhead of maintaining sparse distributions is negligible compared to storing and backpropagating through the LLM, as explained in Equation 23. Further, both memory and computation time for *TROLL* scale only with the vocabulary size, which is constant for most model families. We thus find that *TROLL*'s overhead diminishes as model size increases. Table 5 provides detailed evaluations.

# 7 CONCLUSION

We introduce *TROLL*, a trust-region based policy gradient objective that acts as a drop-in replacement for the popular PPO-clip. *TROLL* is based on a novel, principled, and fully differentiable trust-region projection for discrete distributions. This projection compares two distributions, in our case, the token logit distributions of an old policy used to collect sequences, and a new policy that performs policy gradient updates on these sequences. Since these distributions are prohibitively large for modern vocabulary sizes, we extend the projection to sparse distributions. Here, we only keep a small subset of logits that represent the most likely token predictions, allowing us to realize both data collection and the projection objective using fully sparse operations. We experimentally validate *TROLL* across various model families, model sizes, advantage estimation methods, and math and code generation datasets. *TROLL* significantly and consistently outperforms the PPO-clip objective in terms of sample efficiency and final reward across setups, while only requiring a small overhead that does not scale with model size.

**Limitations and Future Work.** We currently evaluate our method on dense models with up to 14B parameters. In future work, we want to scale *TROLL* to larger models and Mixture-of-Experts architectures. Similarly, it would be interesting to extend *TROLL* to other modalities and tasks, using, for example, vision-language models, where the logit distributions and their projections may behave differently from pure language.

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

## ETHICS STATEMENT

*TROLL* improves the efficiency of LLM finetuning by enabling scalable trust-region optimization. While our experiments focus on mathematical reasoning, the method is broadly applicable to other domains. As with any advance in LLM training, this carries both potential benefits and risks, depending on the context of deployment. We believe that managing and shaping the societal impacts of increasingly powerful LLMs should not be left to individual researchers, organizations, or companies alone, but they must be carefully governed and regulated by sovereign governments and strong democratic institutions.

## REPRODUCIBILITY STATEMENT

All experiments in this paper rely on publicly available pretrained checkpoints. We exclusively use publicly available datasets. While some were modified, we describe these modifications and will release the processed versions upon the deanonymization of the paper. Further information, together with additional hyperparameters and training details, are provided in Appendix C. Our implementation builds on open-source repositories and will be made available after deanonymization.

## ON LLM USAGE

We used LLMs to assist with revising grammar, style, and text flow in this manuscript. In addition, we employed LLMs to support aspects of the implementation and generate visualizations for this manuscript.

## ACKNOWLEDGEMENTS

The authors acknowledge support by the state of Baden-Württemberg through bwHPC, as well as the HoreKa supercomputer funded by the Ministry of Science, Research and the Arts Baden-Württemberg and by the German Federal Ministry of Education and Research.

## A  DERIVATIONS

For each output token $o_t$ the trust region projection layer solves

$$\underset{\pi_\theta(o_t \,|\, \boldsymbol{q}, \boldsymbol{o}_{<t})}{\arg\min} \ \ \mathrm{KL}\left(\pi_\theta(o_t \,|\, \boldsymbol{q}, \boldsymbol{o}_{<t}) \,\|\, \tilde{\pi}_\theta(o_t \,|\, \boldsymbol{q}, \boldsymbol{o}_{<t})\right) \tag{7}$$

$$\text{s.t. } \mathrm{KL}\left(\pi_\theta(o_t \,|\, \boldsymbol{q}, \boldsymbol{o}_{<t}) \,\|\, \pi_{\mathrm{old}}(o_t \,|\, \boldsymbol{q}, \boldsymbol{o}_{<t})\right) < \epsilon \ \text{ and } \ \sum_{o_t}\left[\pi_\theta(o_t \,|\, \boldsymbol{q}, \boldsymbol{o}_{<t})\right] = 1.$$

Here, the first constraint enforces the trust region to the previous distribution $\pi_{\mathrm{old}}(o_t \,|\, \boldsymbol{q}, \boldsymbol{o}_{<t})$ and the second constraint ensures the resulting distribution is properly normalized. We solve the constrained optimization problem using the method of Lagrangian multipliers and start with the primal solution.

### A.1  PRIMAL SOLUTION

To compute the primal solution of this optimization problem, we first set up the Lagrangian function by introducing Lagrangian multipliers $\eta > 0$ and $\lambda$ for the first and second constraint, respectively.

The corresponding Lagrangian is given as

$$\mathcal{L}(\pi_\theta(o_t \,|\, \boldsymbol{q}, \boldsymbol{o}_{<t}), \eta)$$
$$= \mathrm{KL}\left(\pi_\theta(o_t \,|\, \boldsymbol{q}, \boldsymbol{o}_{<t}) \,\|\, \tilde{\pi}_\theta(o_t \,|\, \boldsymbol{q}, \boldsymbol{o}_{<t})\right) - \eta(\epsilon - \mathrm{KL}\left(\pi_\theta(o_t \,|\, \boldsymbol{q}, \boldsymbol{o}_{<t}) \,\|\, \pi_{\mathrm{old}}(o_t \,|\, \boldsymbol{q}, \boldsymbol{o}_{<t})\right)$$
$$\quad - \lambda\left(1 - \sum_{o_t}\left[\pi_\theta(o_t \,|\, \boldsymbol{q}, \boldsymbol{o}_{<t})\right]\right)$$
$$= -(\eta\epsilon + \lambda) + \sum_{o_t}\left[\pi_\theta(o_t \,|\, \boldsymbol{q}, \boldsymbol{o}_{<t})\left(\log\frac{\pi_\theta(o_t \,|\, \boldsymbol{q}, \boldsymbol{o}_{<t})}{\tilde{\pi}_\theta(o_t \,|\, \boldsymbol{q}, \boldsymbol{o}_{<t})} + \eta\log\frac{\pi_\theta(o_t \,|\, \boldsymbol{q}, \boldsymbol{o}_{<t})}{\pi_{\mathrm{old}}(o_t \,|\, \boldsymbol{q}, \boldsymbol{o}_{<t})} + \lambda\right)\right]$$
$$= -(\eta\epsilon + \lambda) + \tag{8}$$
$$\sum_{o_t}\left[\pi_\theta(o_t \,|\, \boldsymbol{q}, \boldsymbol{o}_{<t})\big((\eta+1)\log\pi_\theta(o_t \,|\, \boldsymbol{q}, \boldsymbol{o}_{<t}) - \log\tilde{\pi}_\theta(o_t \,|\, \boldsymbol{q}, \boldsymbol{o}_{<t}) - \eta\log\pi_{\mathrm{old}}(o_t \,|\, \boldsymbol{q}, \boldsymbol{o}_{<t}) + \lambda\big)\right].$$

We can now obtain the optimal primal solution to Equation 7 by taking the derivative of the Lagrangian w.r.t. $\pi_\theta(o_t \,|\, \boldsymbol{q}, \boldsymbol{o}_{<t})$, setting it to 0, and solving for $\pi_\theta(o_t \,|\, \boldsymbol{q}, \boldsymbol{o}_{<t})$. The derivative is given by

$$\frac{\partial\mathcal{L}(\pi_\theta(o_t \,|\, \boldsymbol{q}, \boldsymbol{o}_{<t}), \eta)}{\partial\pi_\theta(o_t \,|\, \boldsymbol{q}, \boldsymbol{o}_{<t})}$$
$$= \sum_o\left[(\eta+1) + (\eta+1)\log\pi_\theta(o_t \,|\, \boldsymbol{q}, \boldsymbol{o}_{<t}) - (\log\tilde{\pi}_\theta(o_t \,|\, \boldsymbol{q}, \boldsymbol{o}_{<t}) + \eta\log\pi_{\mathrm{old}}(o_t \,|\, \boldsymbol{q}, \boldsymbol{o}_{<t})) + \lambda\right].$$

Clearly $\partial\mathcal{L}(\pi_\theta(o_t \,|\, \boldsymbol{q}, \boldsymbol{o}_{<t}), \eta)/\partial\pi_\theta(o_t \,|\, \boldsymbol{q}, \boldsymbol{o}_{<t}) = 0$ if all the individual terms of the sum are $0$. Thus, the problem simplifies to

$$0 = (\eta+1) + (\eta+1)\log\pi_\theta(o_t \,|\, \boldsymbol{q}, \boldsymbol{o}_{<t}) - (\log\tilde{\pi}_\theta(o_t \,|\, \boldsymbol{q}, \boldsymbol{o}_{<t}) + \eta\log\pi_{\mathrm{old}}(o_t \,|\, \boldsymbol{q}, \boldsymbol{o}_{<t})) + \lambda$$

which yields

$$\log\pi_\theta(o_t \,|\, \boldsymbol{q}, \boldsymbol{o}_{<t}) = \frac{\log\tilde{\pi}_\theta(o_t \,|\, \boldsymbol{q}, \boldsymbol{o}_{<t}) + \eta\log\pi_{\mathrm{old}}(o_t \,|\, \boldsymbol{q}, \boldsymbol{o}_{<t})) - (\eta+1+\lambda)}{\eta+1} \tag{9}$$

and thus

$$\pi_\theta(o_t \,|\, \boldsymbol{q}, \boldsymbol{o}_{<t}) = \exp\left(\frac{\log\tilde{\pi}_\theta(o_t \,|\, \boldsymbol{q}, \boldsymbol{o}_{<t}) + \eta\log\pi_{\mathrm{old}}(o_t \,|\, \boldsymbol{q}, \boldsymbol{o}_{<t})}{\eta+1}\right)\exp\left(-\frac{\eta+1+\lambda}{\eta+1}\right)$$
$$\propto \exp\left(\frac{\log\tilde{\pi}_\theta(o_t \,|\, \boldsymbol{q}, \boldsymbol{o}_{<t}) + \eta\log\pi_{\mathrm{old}}(o_t \,|\, \boldsymbol{q}, \boldsymbol{o}_{<t})}{\eta+1}\right) \tag{10}$$

Crucially, this primal solution allows computing a properly normalized distribution $\pi_\theta(o_t \,|\, \boldsymbol{q}, \boldsymbol{o}_{<t})$ without explicitly computing $\lambda$ by replacing the $\exp$ in Equation 10 with a softmax.

## A.2 DUAL SOLUTION

The second step of the Lagrangian multiplier method is to solve the dual problem which finds the optimal dual parameters given the primal solution. To that end, we insert the primal solution from Equation 9 into the Lagrangian (Equation 8). Here most terms cancel out, leading to a dual of the form

$$D(\eta, \lambda) = -\eta\epsilon - \lambda - \eta - 1 = -\eta\epsilon - (\eta+1+\lambda). \tag{11}$$

In a second step towards a practically usable dual, we remove the dependency on $\lambda$ by exploiting the constraint it enforces, i.e., $\sum_{o_t}[\pi_\theta(o_t \,|\, \boldsymbol{q}, \boldsymbol{o}_{<t})] = 1$. Going to log space and again using Equation 9, this property yields

$$0 = \log\sum_{o_t}\left[\pi_\theta(o_t \,|\, \boldsymbol{q}, \boldsymbol{o}_{<t})\right]$$
$$= \log\sum_{o_t}\left[\exp\left(\frac{\log\tilde{\pi}_\theta(o_t \,|\, \boldsymbol{q}, \boldsymbol{o}_{<t}) + \eta\log\pi_{\mathrm{old}}(o_t \,|\, \boldsymbol{q}, \boldsymbol{o}_{<t})}{\eta+1}\right)\exp\left(-\frac{\eta+1+\lambda}{\eta+1}\right)\right]$$
$$= -\frac{\eta+1+\lambda}{\eta+1} + \log\sum_{o_t}\left[\exp\left(\frac{\log\tilde{\pi}_\theta(o_t \,|\, \boldsymbol{q}, \boldsymbol{o}_{<t}) + \eta\log\pi_{\mathrm{old}}(o_t \,|\, \boldsymbol{q}, \boldsymbol{o}_{<t})}{\eta+1}\right)\right]$$

which we can rewrite as

$$\eta + 1 + \lambda = (\eta + 1) \log \sum_{o_t} \left[ \exp \left( \frac{\log \tilde{\pi}_\theta(o_t \,|\, \boldsymbol{q}, \boldsymbol{o}_{<t}) + \eta \log \pi_{\text{old}}(o_t \,|\, \boldsymbol{q}, \boldsymbol{o}_{<t})}{\eta + 1} \right) \right]. \tag{12}$$

Now, inserting Equation 12 into Equation 11 removes the dependency on $\lambda$ leading to

$$D(\eta) = -\eta\epsilon - (\eta + 1) \log \sum_{o_t} \left[ \exp \left( \frac{\log \tilde{\pi}_\theta(o_t \,|\, \boldsymbol{q}, \boldsymbol{o}_{<t}) + \eta \log \pi_{\text{old}}(o_t \,|\, \boldsymbol{q}, \boldsymbol{o}_{<t})}{\eta + 1} \right) \right].$$

Using this dual, we can find the optimal $\eta^*$ by solving

$$\eta^* = \arg\max_\eta D(\eta) \ \ s.t. \ \ \eta \geq 0. \tag{13}$$

We can efficiently optimize this scalar optimization problem using the $n$-ary bracketing method described in Listing 3.

### A.3 GRADIENTS

This trust region projection is trivially differentiable using standard autograd tools, except for the numerical optimization of the dual to find the optimal step size $\eta^*$. Towards differentiating through this optimization problem in closed form, let us first change perspective and no longer consider the distributions $\pi_\theta(o_t \,|\, \boldsymbol{q}, \boldsymbol{o}_{<t})$, $\pi_{\text{old}}(o_t \,|\, \boldsymbol{q}, \boldsymbol{o}_{<t})$, and $\tilde{\pi}_\theta(o_t \,|\, \boldsymbol{q}, \boldsymbol{o}_{<t})$ directly but vectors $q$, $q_{\text{old}}^{(\log)}$, and $\tilde{q}^{(\log)}$. Here $q$ corresponds to the probabilities of $\pi_\theta(o_t \,|\, \boldsymbol{q}, \boldsymbol{o}_{<t})$ while $q_{\text{old}}^{(\log)}$ and $\tilde{q}^{(\log)}$ denote to the normalized logits of $\pi_{\text{old}}(o_t \,|\, \boldsymbol{q}, \boldsymbol{o}_{<t})$ and $\tilde{\pi}_\theta(o_t \,|\, \boldsymbol{q}, \boldsymbol{o}_{<t})$. We further assume all 3 vectors are normalized, i.e.,

$$\sum q = 1, \quad \sum \exp q_{\text{old}}^{(\log)} = 1 \quad \text{and} \quad \sum \exp \tilde{q}^{(\log)} = 1.$$

While this notation may seem slightly unintuitive at first, it simplifies the following derivations. As we assume the $\pi_{\text{old}}(o_t \,|\, \boldsymbol{q}, \boldsymbol{o}_{<t})$ and consequently $q_{\text{old}}^{(\log)}$ are constant, the only gradient we are interested in is $\dfrac{\partial \eta^*}{\partial \tilde{q}^{(\log)}}$, i.e., how the original LLM's output influences the optimal step size $\eta^*$.

Since we do not have an analytical form for the optimal step size $\eta^*$ but only the result of the numerical optimization, we need to introduce further analytical properties. Using the implicit differentiation (Dontchev & Rockafellar, 2009) and differentiable matrix calculus (Magnus & Neudecker, 1989) techniques introduced to deep learning by OptNet (Amos & Kolter, 2017), we start by writing out the Karush–Kuhn–Tucker (KKT) conditions (Karush, 1939) of the dual in Equation 13 for the optimum at $\eta^*$. Denoting the Lagrangian multiplier corresponding to the $\eta \geq 0$ constraint by $\mu$ and realizing that $\nabla D(\eta) = \epsilon - \text{KL}\left(\pi_\theta(o_t \,|\, \boldsymbol{q}, \boldsymbol{o}_{<t}) \,\|\, \pi_{\text{old}}(o_t \,|\, \boldsymbol{q}, \boldsymbol{o}_{<t})\right) = \epsilon - q^T(\log q - q_{\text{old}}^{(\log)})$, those are given by

$$\underbrace{\nabla D(\eta^*) + \mu \nabla(-\eta^*) = \epsilon - q^T(\log q - q_{\text{old}}^{(\log)}) - \mu = 0}_{\text{Stationarity}} \quad \text{and} \quad \underbrace{\mu(-\eta^*) = 0}_{\text{Complementary Slackness}} \ .$$

As there is no equality constraint in Equation 13, primal feasibility is given by default. We can now take the total differentials around these conditions, which are given by

$$0 = d\left(\epsilon - q^T(\log q - q_{\text{old}}^{(\log)}) - \mu\right) = -d\left(q^T(\log q - q_{\text{old}}^{(\log)})\right) - d\mu \tag{14}$$

$$0 = d(\mu(-\eta^*)) = d\mu(-\eta^*) + \mu(-d\eta^*), \tag{15}$$

where $d\epsilon$ vanishes as it is constant. Before proceeding, we need to rewrite the KL term $d\left(q^T(\log q - q_{\text{old}}^{(\log)})\right)$ in terms of $\tilde{q}^{(\log)}$ and simplify. First, we have

$$d\left(q^T(\log q - q_{\text{old}}^{(\log)})\right) = (1 + \log q - q_{\text{old}}^{(\log)})^T dq \tag{16}$$

and need to continue with the differential $dq$. Again using the primal solution Equation 10, we get

$$q = \text{softmax}\left(\frac{\eta^* q_{\text{old}}^{(\log)} + \tilde{q}^{(\log)}}{\eta^* + 1}\right). \tag{17}$$

Assuming the old logits are a constant, we can write the corresponding differential as

$$dq = \frac{\partial q}{\partial \tilde{q}^{(\log)}} d\tilde{q}^{(\log)} + \frac{\partial q}{\partial \eta^*} d\eta^*.$$

Inserting this term into Equation 16 and the plugging the result into Equation 14 yields

$$- \left(1 + \log q - q_{\text{old}}^{(\log)}\right)^T \frac{\partial q}{\partial \eta^*} d\eta^* - d\mu = \left(1 + \log q - q_{\text{old}}^{(\log)}\right) \frac{\partial q}{\partial \tilde{q}^{(\log)}} d\tilde{q}^{(\log)} \tag{18}$$

$$-\mu d\eta^* - \eta^* d\mu = 0, \tag{19}$$

which we can use to compute the desired gradient $\dfrac{\partial \tilde{q}^{(\log)}}{\partial \eta^*}$. To this end, we consider two separate cases. First, if the original KL trust region is not violated, then $\eta^* = 0$ and $\mu > 0$. In this case, Equation 19 directly yields that $d\eta^* = 0$ and thus the entire gradient $\dfrac{\partial \eta^*}{\partial \tilde{q}^{(\log)}}$ is zero. Second, the original KL trust region constraint is active and thus $\eta^* > 0$ and $\mu = 0$. In this case Equation 19 gives $d\mu = 0$ which simplifies Equation 18. Reordering the remaining terms gives the required gradient

$$\frac{\partial \eta^*}{\partial \tilde{q}^{(\log)}} = \frac{1}{-(1 + \log q - q_{\text{old}}^{(\log)})^T \frac{\partial q}{\partial \eta^*}} (1 + \log q - q_{\text{old}}^{(\log)})^T \frac{\partial q}{\partial \tilde{q}^{(\log)}}$$

The required partial derivatives can be obtained from Equation 17 as

$$\frac{\partial q}{\partial \tilde{q}^{(\log)}} = \frac{1}{\eta^* + 1}(\mathrm{D}(q) - qq^T) \quad \text{and} \quad \frac{\partial q}{\partial \eta^*} = \frac{1}{(\eta + 1)^2}(\mathrm{D}(q) - qq^T)(q_{\text{old}}^{(\log)} - \tilde{q}^{(\log)}),$$

where $D(q)$ denotes a diagonal matrix with the entries of $q$ on the diagonal.

Crucially, for practical purposes, we never need to explicitly materialize the matrices in the partial derivatives. The resulting backward introduces negligible computational and memory overhead and. Listing 2 shows that, in the non-sparsified case, the backward can be written in less than ten lines of python code.

## A.4 SPARSIFICATION

**Theorem A.1.** *For any pair of logits $o_t^{(1)}$ and $o_t^{(2)}$, with $\tilde{\pi}(o_t^{(1)} \mid \boldsymbol{q}, \boldsymbol{o}_{<t}) \geq \tilde{\pi}(o_t^{(2)} \mid \boldsymbol{q}, \boldsymbol{o}_{<t})$ w.l.o.g., the logit-wise terms that sum to the KL are equally ordered*

$$\tilde{\pi}(o_t^{(1)} \mid \boldsymbol{q}, \boldsymbol{o}_{<t}) \log \frac{\tilde{\pi}(o_t^{(1)} \mid \boldsymbol{q}, \boldsymbol{o}_{<t})}{\pi_{old}(o_t^{(1)} \mid \boldsymbol{q}, \boldsymbol{o}_{<t})} \geq \tilde{\pi}(o_t^{(2)} \mid \boldsymbol{q}, \boldsymbol{o}_{<t}) \log \frac{\tilde{\pi}(o_t^{(2)} \mid \boldsymbol{q}, \boldsymbol{o}_{<t})}{\pi_{old}(o_t^{(2)} \mid \boldsymbol{q}, \boldsymbol{o}_{<t})} \tag{20}$$

*iff $e^\kappa \geq \gamma$, where $\kappa = \frac{\tilde{\pi}(o_t^{(1)} \mid \boldsymbol{q}, \boldsymbol{o}_{<t})}{\tilde{\pi}(o_t^{(2)} \mid \boldsymbol{q}, \boldsymbol{o}_{<t})}$ is the current probability ratio of the pair and $\gamma$ in $\frac{\pi_{old}(o_t^{(1)} \mid \boldsymbol{q}, \boldsymbol{o}_{<t})}{\pi_{old}(o_t^{(2)} \mid \boldsymbol{q}, \boldsymbol{o}_{<t})} = \gamma\kappa$ gives the multiplier of the old ratio.*

*Proof.* Rewrite $\tilde{\pi}(o_t^{(1)} \mid \boldsymbol{q}, \boldsymbol{o}_{<t}) \geq \tilde{\pi}(o_t^{(2)} \mid \boldsymbol{q}, \boldsymbol{o}_{<t})$ as $p(x_1) = \kappa \cdot p(x_2)$ for $\kappa \geq 1$ using $p(x_i) = \tilde{\pi}(o_t^{(i)} \mid \boldsymbol{q}, \boldsymbol{o}_{<t})$ for clarity and similarly replace $q(x_i) = \pi_{old}(o_t^{(i)} \mid \boldsymbol{q}, \boldsymbol{o}_{<t})$. Then compare the contributions of $x_1$ and $x_2$ to the KL divergence

$$\kappa p(x_2) \log \frac{\kappa p(x_2)}{q(x_1)} \geq p(x_2) \log \frac{p(x_2)}{q(x_2)}$$

$$\kappa \log k \geq \log \frac{q(x_1)}{q(x_2)}$$

and substitute $\frac{q(x_1)}{q(x_2)} =: \gamma \frac{p(x_1)}{p(x_2)} = \gamma\kappa$

$$e^\kappa \kappa \geq \gamma \frac{p(x_1)}{p(x_2)}$$

$$e^\kappa \geq \gamma.$$

$\square$

Here, the assumption that the relative likelihood $\kappa$ of $o_t^{(1)}$ and $o_t^{(2)}$ was not exponentially larger before usually holds in practice, as the token distributions are pushed farther from uniform during training (Cui et al., 2025b).

**Definition A.1.** For any subset $\mathcal{S}$ of the possible tokens, we define $p_\mathcal{S}$, or just $p'$ when the mask is clear, as the *sparsed* distribution. For tokens not in $\mathcal{S}$, it has default probability $p_d$ and the same probability as $p$ for all others up to the renormalization constant.

$$p_\mathcal{S}(x) = p'(x) := \begin{cases} \gamma p(x), & \text{for } x \in \mathcal{S} \\ p_d, & \text{else} \end{cases}, \qquad \gamma = \frac{1 - (|\mathcal{V}| - |\mathcal{S}|) \cdot p_d}{\sum_{x \in \mathcal{S}} p(x)}. \tag{21}$$

The renormalization factor $\gamma$ accounts for the previous total mass $\sum_{x \in \mathcal{S}} p(x)$ of the kept tokens and new mass $(|\mathcal{V}| - |\mathcal{S}|) \cdot p_d$ of the dropped tokens.

In the case of equal sparsification masks for distributions $p, q$, we can prove a practically tight upper bound for the true divergence KL $(p \parallel q)$ in terms of the sparse divergence KL $(p' \parallel q')$.

**Theorem A.2.** *Let $p, q$ be categorical distributions over the vocabulary $|\mathcal{V}|$ with identical top-$k$ logits,* $\text{topk(p)} = \text{topk(q)}$ *and equal total probability* $\sum_{x \in \text{topk(p)}} p(x) = \sum_{x \in \text{topk(q)}} q(x) = 1 - \delta$. *Then the sparsed distributions $p', q'$ with density*

$$p'(x) := \begin{cases} \gamma p(x), & \text{for } x \in \text{topk(p)} \\ p_d, & \text{else} \end{cases}, \quad q'(x_i) := \begin{cases} \gamma q(x), & \text{for } x \in \text{topk(q)} \\ p_d, & \text{else} \end{cases},$$

*and normalization constant $\gamma(\delta, k, |\mathcal{V}|, p_d) \approx 1$ follow the inequality*

$$KL\,(p \parallel q) \le \gamma^{-1} KL\,(p' \parallel q') + \delta \log \frac{\delta}{q_{\min}},$$

*where $q_{\min} = \arg\min_x q(x)$.*

*Proof.* Rename the logits in descending order of probability under p, such that $p(x_0) \ge p(x_1) \ge \cdots \ge p(x_{|\mathcal{V}|-1})$. Assume there is $k < |\mathcal{V}|$ such that the largest $k$ logits of both $p$ and $q$ have exactly the total probability mass $\sum_{i=0}^{k} p(x_i) = \sum_{i=0}^{k-1} q(x_i) = 1 - \delta$ and the subset of largest logits is identical. Every nondegenerate distribution has $q_{\min} \le q(x_i)$ and all $p(x_i) \le \delta$ for $i \ge k$, as the total mass could otherwise not be $1 - \delta$. So split the sum over logits in the KL divergence and apply both inequalities

$$\begin{aligned}
\text{KL}\,(p \parallel q) &= \sum_{i=0}^{k-1} p(x_i) \log \frac{p(x_i)}{q(x_i)} + \sum_{i=k}^{|\mathcal{V}|-1} p(x_i) \log \frac{p(x_i)}{q(x_i)} \\
&\le \sum_{i=0}^{k-1} p(x_i) \log \frac{p(x_i)}{q(x_i)} + \sum_{i=k}^{|\mathcal{V}|-1} p(x_i) \log \frac{\delta}{q(x_i)} \\
&\le \sum_{i=0}^{k-1} p(x_i) \log \frac{p(x_i)}{q(x_i)} + \sum_{i=k}^{|\mathcal{V}|-1} p(x_i) \log \frac{\delta}{q_{\min}} \\
&= \sum_{i=0}^{k-1} p(x_i) \log \frac{p(x_i)}{q(x_i)} + \log \frac{\delta}{q_{\min}} \underbrace{\sum_{i=k}^{|\mathcal{V}|-1} p(x_i)}_{= \delta} \\
&= \sum_{i=0}^{k-1} p(x_i) \log \frac{p(x_i)}{q(x_i)} + \delta \log \frac{\delta}{q_{\min}}.
\end{aligned}$$

Replace p, and analogously q, with their sparsed version as defined in Definition A.1,

$$p'(x_i) := \begin{cases} \gamma p(x_i), & \text{for } i < k \\ p_d, & \text{for } i \ge k \end{cases}, \tag{22}$$

```
1  def TROLLProjection(log_target_prob, log_ref_prob, bound):
2      kl_div = (log_target_prob.exp() * (log_target_prob - log_ref_prob)).
           sum(dim=-1)
3      needs_projection = kl_div >= bound # only projects where necessary
4      # ... masking of needed tokens
5      # solve dual problem, i.e. find $\eta^*$
6      opt_eta = DualSolver(log_target_prob, log_ref_prob, bound)
7      primal_unnormalized = (opt_eta * log_ref_prob + log_target_prob) / (
           opt_eta + 1)
8      primal = inner.log_softmax(dim=-1)
9      # ... combine masked unprojected and primal logits into one
10     return projected_logits
```

Listing 1: *TROLL*'s differentiable projection only calls a differentiable dual solver and otherwise uses standard autodiff operations.

where $\gamma = \frac{1-(|\mathcal{V}|-k)\cdot p_d}{(1-\delta)}$ renormalizes the $(1-\delta)$ mass of the selected tokens to account for the default mass $(|\mathcal{V}| - k) \cdot p_d$ of the sparsified tokens. Multiplying with ones and adding a zero to the KL bound yields the relation to the sparse KL,

$$
\begin{aligned}
\mathrm{KL}\left(p \parallel q\right) &\le \sum_{i=0}^{k-1} \frac{\gamma}{\gamma} p(x_i) \log \frac{\gamma p(x_i)}{\gamma q(x_i)} + \delta \log \frac{\delta}{q_{\min}} + \gamma^{-1} \underbrace{\sum_{i=k}^{|\mathcal{V}|-1} p'(x_i) \log \frac{p_d}{p_d}}_{=0} \\
&= \gamma^{-1} \sum_{i=0}^{k-1} p'(x_i) \log \frac{p'(x_i)}{q'(x_i)} + \delta \log \frac{\delta}{q_{\min}} + \gamma^{-1} \sum_{i=k}^{|\mathcal{V}|-1} p'(x_i) \log \frac{p'(x_i)}{q'(x_i)} \\
&= \gamma^{-1} \mathrm{KL}\left(p' \parallel q'\right) + \delta \log \frac{\delta}{q_{\min}}.
\end{aligned}
$$

Assuming that $q$'s probabilities can be represented by normal single precision IEEE-754 numbers, $q_{\min} > 1.17549 \cdot 10^{-38}$, and $k \ll |\mathcal{V}|$, e.g. $k = 256$ of vocab size $|\mathcal{V}| = 151936$ while using threshold $\delta = 10^{-5}$ and default mass $p_d = 10^{-12}$, the sparse KL approximation,

$$
\begin{aligned}
\mathrm{KL}\left(p \parallel q\right) &\le \frac{(1-\delta)}{1 - (|\mathcal{V}| - k) \cdot p_d} \mathrm{KL}\left(p' \parallel q'\right) + \delta \log \frac{\delta}{q_{\min}} \\
&= \frac{0.99999}{1 - 151680 \cdot 10^{-12}} \mathrm{KL}\left(p' \parallel q'\right) + 10^{-5} \log \frac{10^{-5}}{1.17549 \cdot 10^{-38}} \\
&\le 0.99999015168 \cdot \mathrm{KL}\left(p' \parallel q'\right) + 0.00075823623,
\end{aligned}
$$

is accurate enough for limiting the true divergence to values on the order of $0.05$ as

$$
\begin{aligned}
\mathrm{KL}\left(p \parallel q\right) &\le 0.99999015168 \cdot \mathrm{KL}\left(p' \parallel q'\right) + 0.00075823623 \\
&\le 0.99999015168 \cdot 0.05 + 0.00075823623 \\
\mathrm{KL}\left(p \parallel q\right) &\le 0.050757743814.
\end{aligned}
$$

$\square$

## B   IMPLEMENTATION

While the theoretical derivation of the differentiable trust region projection can look complex, the final implementation is fairly straightforward. We give PyTorch-adjacent pseudocode for the dense variant of the primal (Listing 1) and dual (Listing 2) in the following. Note that the sparse implementation mostly differs in the usage of a custom sparse tensor class that maintains a default probability for the implicit entries. While this requires additional care in terms of indexing and allows for optimizations of, e.g., the KL computation, the general logic remains unchanged. Listing 3, shows our $n$-ary bracketing method to optimize the dual.

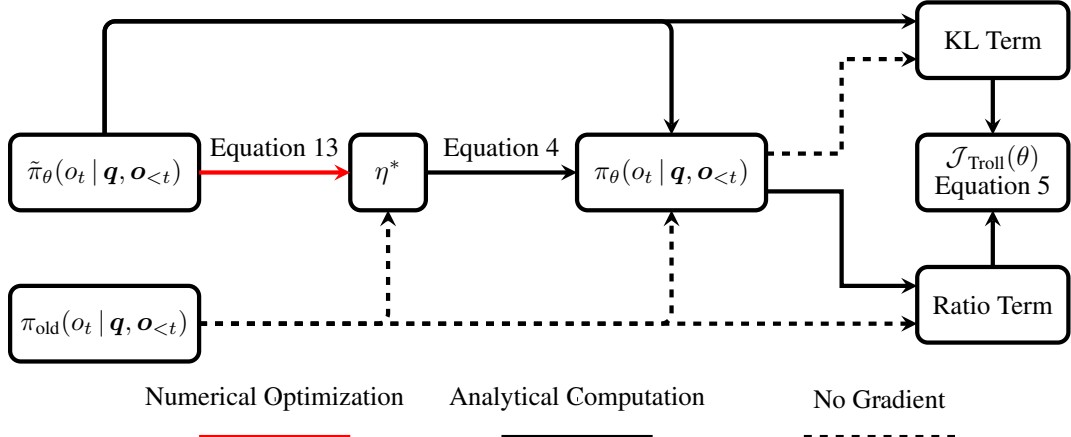

Figure 6: Compute Graph from the LLM output $\tilde{\pi}_\theta(o_t \mid \boldsymbol{q}, \boldsymbol{o}_{<t})$ to the RL objective $\mathcal{J}_{\text{Troll}}(\theta)$. First, the optimal step size $\eta^*$ is computed for each token. Here, we use the fact that the step size for tokens that do not violate the trust region is trivially 0. For the tokens that do violate the trust region, we need to optimize a 1-D convex optimization problem to compute $\eta^*$. Next, the projection computes the optimal distribution within the trust region, $\pi_\theta(o_t \mid \boldsymbol{q}, \boldsymbol{o}_{<t})$, which is then used in the objective. This objective combines the standard policy ratio term from PPO with a KL term to regress $\tilde{\pi}_\theta(o_t \mid \boldsymbol{q}, \boldsymbol{o}_{<t})$ towards $\pi_\theta(o_t \mid \boldsymbol{q}, \boldsymbol{o}_{<t})$.

```python
def DualSolver.forward(log_target_prob, log_ref_prob, bound):
    # define objective in terms of log eta (such that eta > 0)
    opt_log_eta = optimize1d(
            lambda log_eta: dual(log_eta, ...),
            # ... bounds and termination config
        )
    # ... save for backward
    return opt_log_eta.exp()

def dual(log_eta, bound, log_target_prob, log_ref_prob)
    eta = log_eta.exp()
    inner = (log_target_prob + eta * log_ref_prob) / (eta + 1)
    inner_lse = logsumexp(inner, axis=-1)
    # negative of objective, since we minimize
    return eta *  bound + (eta + 1) * inner_lse

def DualSolver.backward(grad_output):
    # ... recompute primal = ... as in TROLLProjection
    one_plus_logratio = 1 + primal.log() - log_ref_prob
    # compute one_plus_logratio.T @ dprimal_dlog_output implicitly
    numerator = primal * (one_plus_logratio - vecdot(primal,
        one_plus_logratio).unsqueeze(-1) / (opt_eta + 1)
    # compute  dprimal_dopt_eta implicitly
    diff = log_ref_prob - log_target_prob
    dprimal_dopt_eta = primal * (diff - vecdot(primal, diff).unsqueeze
        (-1) / (opt_eta + 1)**2
    return grad_output * (numerator / -vecdot(one_plus_logratio,
        dprimal_dopt_eta))
```

Listing 2: Custom forward and backward code for *TROLL*'s dual solver.

```python
class Optimizer1D:

    def batched_linspace(lower, upper, num_points):
        # Batched linspace: lower and upper are (batch_size, 1), returns
            (batch_size, num_points)

        steps = linspace(0, 1, num_points)
        return lower + (upper - lower) * steps

    def _opt_step(func, x, lower, upper):
        batch_size, num_points = x.shape
        # batched evaluation of all points
        y = func(x)
        # select min index for each batch element
        min_idx = argmin(y, dim=1)

        # take left and right point
        l_idx = min_idx - 1
        u_idx = min_idx + 1
        l_tmp = x[arange(batch_size), clamp(l_idx, 0, num_points - 1)]
        u_tmp = x[arange(batch_size), clamp(u_idx, 0, num_points - 1)]
        new_lower = where(l_idx < 0, lower), l_tmp)
        new_upper = where(u_idx >= num_points, upper, u_tmp)
        return new_lower, new_upper

    def optimize(func, lower, upper, num_points, max_steps, x_threshold):
        # batched, parallel, gradient-free, optimization of a 1D function

        l, u = lower, upper
        # refine lower and upper until convergence
        for step in range(max_steps):
            x = Optimizer1D.batched_linspace(l, u, num_points + 2)
            x = x[:, 1:-1]

            l, u = Optimizer1D._opt_step(func, x, l, u)

            if ((l - u) < x_threshold).abs().all():
                break

        x = (l + u) / 2

        return x
```

Listing 3: N-ary Bracketing Search.

| Model | Link |
|-------|------|
| Qwen3-0.6B | https://huggingface.co/Qwen/Qwen3-0.6B |
| Qwen3-1.7B | https://huggingface.co/Qwen/Qwen3-1.7B |
| Qwen3-4B | https://huggingface.co/Qwen/Qwen3-4B |
| Qwen3-8B | https://huggingface.co/Qwen/Qwen3-8B |
| Qwen3-14B | https://huggingface.co/Qwen/Qwen3-14B |
| Qwen2.5-0.5B-Instruct | https://huggingface.co/Qwen/Qwen2.5-0.5B-Instruct |
| Qwen2.5-1.5B-Instruct | https://huggingface.co/Qwen/Qwen2.5-1.5B-Instruct |
| Qwen2.5-3B-Instruct | https://huggingface.co/Qwen/Qwen2.5-3B-Instruct |
| Qwen2.5-7B-Instruct | https://huggingface.co/Qwen/Qwen2.5-7B-Instruct |
| Llama-3.1.8B | https://huggingface.co/meta-llama/Llama-3.1-8B |
| Llama-3.1.8B-Instruct | https://huggingface.co/meta-llama/Llama-3.1-8B-Instruct |
| Llama-3.2-3B | https://huggingface.co/meta-llama/Llama-3.2-3B |
| LLama-3.2-3B-Instruct | https://huggingface.co/meta-llama/Llama-3.2-3B-Instruct |
| FineMath-Llama 3B | https://huggingface.co/HuggingFaceTB/FineMath-Llama-3B |
| Apertus-8B | https://huggingface.co/swiss-ai/Apertus-8B-2509 |
| Apertus-8B-Instruct | https://huggingface.co/swiss-ai/Apertus-8B-Instruct-2509 |
| SmolLM3-3B | https://huggingface.co/HuggingFaceTB/SmolLM3-3B |

Table 2: Model checkpoints used as starting points for finetuning throughout this work.

## C    EXPERIMENTAL SETUP

### C.1    MODELS

Table 2 lists all model checkpoints used in this work. They are publicly available and can be downloaded under the provided links.

We used the thinking mode for the models from the Qwen3-Family. For the non-instruct versions of Llama-3.1, Llama-3.2, and Apertus, we used the chat templates from the respective instruct versions.

### C.2    DATASETS

For all math datasets, sequence-level binary rewards are computed by parsing the LLM output through a regular expression, matching against a ground truth answer. For code generation, we evaluate each answer using the provided test cases. The reward is then computed as the fraction of successful tests. If the execution of a test case exceeds a timeout of ten seconds, the evaluation is terminated and zero reward is given.

**DAPO-Math.** We build `DAPO` Train and `DAPO` Eval on the version of the DAPO-Math dataset provided by Cui et al. (2025b)[4] From their training set, we set aside 1024 samples as an in-domain validation set (`DAPO` Eval), leaving 16,893 samples for `DAPO` Train. For broader out-of-distribution evaluation, we again follow Cui et al. (2025b) and use a benchmark suite, we refer to as `Math-Eval`, consisting of MATH500 (Hendrycks et al., 2021), AMC, AIME2024 (Li et al., 2024b), AIME 2025, OMNI-MATH (Gao et al., 2025), OlympiadBench (He et al., 2024), and Minerva (Lewkowycz et al., 2022). We again build the data provided by Cui et al. (2025b) and also follow their protocol by computing the mean over 32 responses for the small but hard AMC, AIME2024, and AIME2025 datasets while only considering a single response for the other sets.

Finally, we ensure all 3 datasets have the same system preprompt, which we provide in Listing 4, and include correct and identical instructions for answer formatting.

---

[4]Their original datasets can be downloaded under https://github.com/PRIME-RL/Entropy-Mechanism-of-RL.

```
Your task is to follow a systematic, thorough reasoning process before
providing the final solution. This involves analyzing, summarizing,
exploring, reassessing, and refining your thought process through
multiple iterations. Structure your response into two sections: Thought
and Solution. In the Thought section, present your reasoning using the
format: "<think> {thoughts} </think>".
```

Listing 4: System Prompt for `DAPO`-Train, `DAPO`-Eval, and `Math`-Eval

| Hyperparameter | Variable | Value |
|---|---|---|
| Trust Region Size | $\epsilon$ | 0.05 |
| KL Regression Factor | $\alpha$ | 1.0 |
| Sparsity Remaining Mass | $1-\delta$ | 0.99999 |
| Max. Sparse Tokens | $K$ | 64 |
| Chunk Size | | 1024 |
| Clip Value | $\epsilon_{\text{ppo}}$ | 0.2 |
| Learning Rate | | $10^{-6}$ |
| Gradient Max Norm | | 1.0 |
| Weight Decay | | 0.0 |
| Learning Rate-Schedule | | constant |
| Learning Rate Critic (PPO only) | | $10^{-5}$ |
| Weight Decay Critic (PPO only) | | 0.01 |
| Sampler Per Query | | 8 |
| Batch Size | | 32 |
| Batches Per Step | | 8 |

Table 3: Hyperparameters. We use these parameters for all experiments unless mentioned otherwise.

**GSM8K.** We use the publicly available train and validation sets of the `GSM8K` Dataset (Hendrycks et al., 2021)[5] without further modifications.

**Eurus-Math.** We use the publicly available train and validation sets of the `Eurus-2-RL-Dataset` (Cui et al., 2025a)[6],which is a subset of NuminaMath-CoT (Li et al., 2024b). We filter for math questions, resulting in $455\,261$ train and $1\,024$ evaluation questions, and refer to the resulting dataset as `Eurus-Math`.

**Eurus-Code.** We use the same `Eurus-2-RL-Dataset` (Cui et al., 2025a) for code generation by filtering for code questions, resulting in $25\,276$ train and $1\,024$ evaluation questions. We call this subset `Eurus-Code`. Each includes multiple test cases of inputs and expected outputs after running the parsed python code within a sandbox. We use the PRIME reward manager of `verl`[7] which evaluates up to the first ten test cases in the `SandboxFusion` sandbox[8].

The dataset consist of tasks from APPS (Hendrycks et al., 2021), CodeContests (Li et al., 2022), TACO (Li et al., 2023) and Codeforces[9]. For the evaluation, we compute the pass@1 scores for each of the four benchmarks and average the results. The validation split has 142, 377, 382 and 123 predefined questions for APPS, CodeContests, TACO, and Codeforces, respectively. The train data is split 13.7%, 38.1%, 37.9% and 10.3%, respectively, such that the evaluation overweights APPS and Codeforces. Empirically, the models seem to perform better on Codeforces but worse on TACO, such that the evaluation success rates are slightly higher.

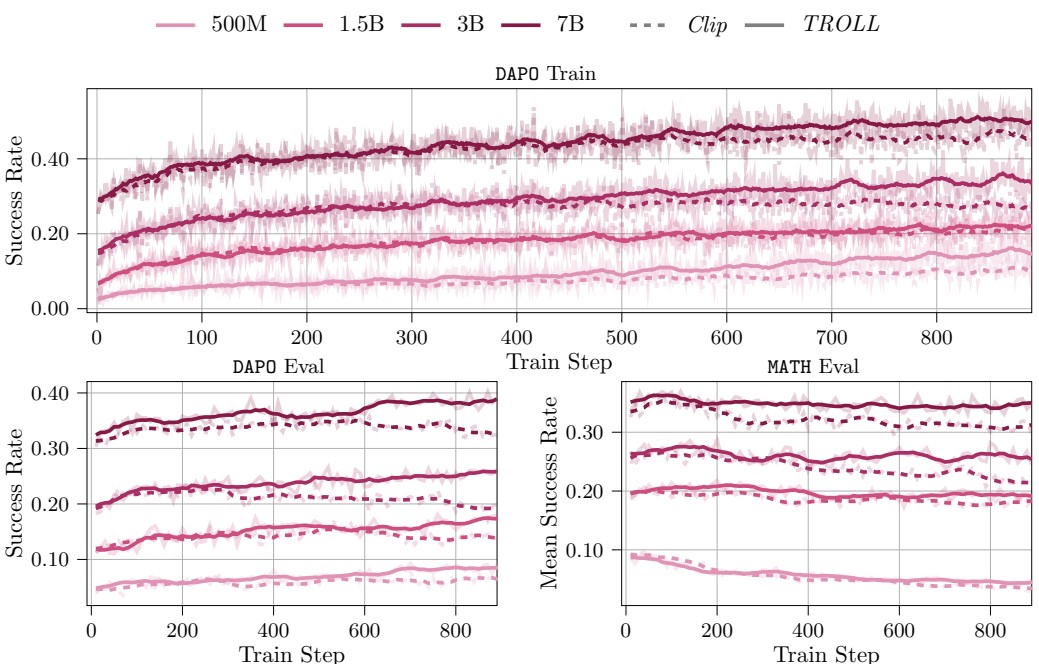

Figure 7: Performance of *TROLL* and the *Clip* objective across Qwen2.5-Instruct models with 500M to 7B parameters trained with GRPO on `DAPO`. As in Figure 3, *TROLL* yields more sample-efficient training and higher rewards at convergence. These improvements extend both to evaluation on in-distribution questions and to generalization on out-of-distribution test datasets. Smoothed values are shown in full opacity, with original curves in the background.

## C.3 TRAINING SETUP

We provide hyperparameters for our training setup in Table 3. We maintain consistent hyperparameters across all experiments, except for Appendix E.1, where we always vary exactly one parameter.

## C.4 HARDWARE

We train on clusters with Nvidia A100, H100, and H200 nodes, each equipped with 4 GPUs. For the Qwen3-14B, Qwen3-8B and Qwen2.5-7B-Instruct experiments in Section 5, we use H200s. For all other experiments, we use either H100 or A100 nodes, depending on model size. We train most experiments for up to 2 days, and extend some experiments on `DAPO` to up to 4 days to show algorithm convergence. We always train *Clip* and *TROLL* on identical hardware to ensure a fair comparison.

# D ADDITIONAL RESULTS

## D.1 QWEN ON `DAPO`

Figure 7 extends the setup of the math evaluation of Figure 3 to Qwen2.5-Instruct models. Similarly to the Qwen3 results, *TROLL* consistently improves over the *Clip* objective for each model size. We further find that, generally, most Qwen2.5 models slightly overfit on the training data, although this effect is less pronounced for *TROLL*.

---

[5] `https://huggingface.co/datasets/openai/gsm8k`

[6] `https://huggingface.co/datasets/PRIME-RL/Eurus-2-RL-Data`

[7] `https://github.com/volcengine/verl`

[8] `https://github.com/bytedance/SandboxFusion`

[9] `https://huggingface.co/datasets/MatrixStudio/Codeforces-Python-Submissions`

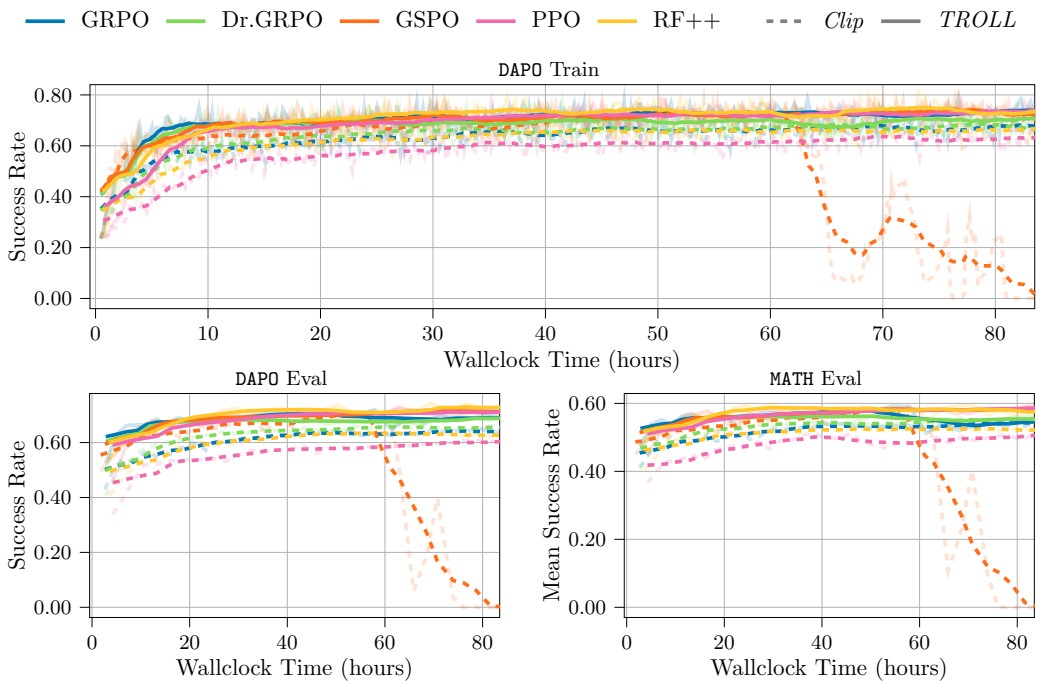

Figure 8: *TROLL* and *Clip* success rates for Qwen3-8B trained with GRPO, Dr.GRPO, GSPO, PPO and RF++ on training data (**top**), in-domain evaluation (**bottom left**) and out-of-domain evaluation (**bottom right**). Smoothed values are shown in full opacity, with original curves in the background. *TROLL* improves over the *Clip* objective for all methods. For GSPO, *Clip* eventually diverges, leading to $0.00\%$ success rate on all metrics, while *TROLL*'s optimization stays stable.

Figure 8 and Figure 9 show complete training and evaluation curves for Table 1. We find that *TROLL* improves training success rates over *Clip* for both models and across methods, to the point where Qwen3 GRPO and Dr.GRPO start to slightly overfit on the out-of-distribution MATH evaluation. Interestingly, while *Clip* leads to unstable performance and eventual divergence for GSPO for both Qwen2.5 and Qwen3, *TROLL*'s token-level trust region optimization remains stable.

## D.2 ADDITIONAL MODELS

Figure 10 and Figure 11 show success rates for different 3B and 8B models, respectively. We find that *TROLL* causes some models, such as Finemath-3B, Llama3.2-3B and Llama3.1-8B to receive a training signal in significantly fewer steps. Other models, such as Apertus-8B show more stable performance when trained with *TROLL*. Finally, for models that work well with the *Clip* objective, using *TROLL* generally yields some performance benefit even though the success rates on GSM8K are almost saturated.

## D.3 QWEN3 ON EURUS-MATH AND GSM8K

We additionally evaluate different Qwen3 model sizes on GSM8K in Figure 12, finding that most models quickly saturate on this comparatively easy task. Nevertheless, using *TROLL* instead of *Clip* generally provides a small boost in performance across model sizes. Similarly, Figure 13 shows that *TROLL* leads to improvements for Qwen3-8B trained with GRPO on Eurus-Math.

## D.4 ADDITIONAL TRAINING ALGORITHMS

Finally, we evaluate Qwen3-8B using BAPO (Xi et al., 2025) as an adaptive clipping heuristic that acts as an alternative to regular clipping, and GPG (Chu et al., 2025), which is a clipping-free policy

| Method | | AIME24 | AIME25 | AMC | MATH | Omni-Math | Olympiad | Minerva |
|--------|------|--------|--------|-----|------|-----------|----------|---------|
| | | | | Qwen2.5-7B-Instruct | | | | |
| GRPO | *Clip* | 0.066 | 0.075 | 0.535 | 0.683 | 0.239 | 0.286 | 0.304 |
| | *TROLL* | 0.168 | 0.129 | 0.587 | 0.712 | 0.254 | 0.317 | 0.284 |
| Dr.GRPO | *Clip* | 0.103 | 0.067 | 0.560 | 0.662 | 0.242 | 0.288 | 0.295 |
| | *TROLL* | 0.168 | 0.135 | 0.605 | 0.706 | 0.259 | 0.320 | 0.317 |
| PPO | *Clip* | 0.092 | 0.064 | 0.503 | 0.706 | 0.251 | 0.316 | 0.299 |
| | *TROLL* | 0.162 | 0.093 | 0.547 | 0.734 | 0.258 | 0.320 | 0.332 |
| GSPO | *Clip* | 0.026 | 0.002 | 0.188 | 0.344 | 0.106 | 0.102 | 0.120 |
| | *TROLL* | 0.159 | 0.076 | 0.531 | 0.699 | 0.257 | 0.297 | 0.310 |
| RF++ | *Clip* | 0.096 | 0.033 | 0.556 | 0.666 | 0.239 | 0.291 | 0.299 |
| | *TROLL* | 0.166 | 0.102 | 0.552 | 0.687 | 0.263 | 0.320 | 0.319 |
| | | | | Qwen3-8B | | | | |
| GRPO | *Clip* | 0.439 | 0.293 | 0.720 | 0.889 | 0.465 | 0.547 | 0.431 |
| | *TROLL* | 0.547 | 0.353 | 0.790 | 0.812 | 0.465 | 0.497 | 0.391 |
| Dr.GRPO | *Clip* | 0.458 | 0.305 | 0.743 | 0.891 | 0.477 | 0.547 | 0.425 |
| | *TROLL* | 0.447 | 0.337 | 0.769 | 0.880 | 0.466 | 0.522 | 0.403 |
| PPO | *Clip* | 0.380 | 0.234 | 0.694 | 0.874 | 0.439 | 0.531 | 0.405 |
| | *TROLL* | 0.524 | 0.408 | 0.780 | 0.910 | 0.521 | 0.567 | 0.425 |
| GSPO | *Clip* | 0.000 | 0.000 | 0.000 | 0.000 | 0.000 | 0.000 | 0.000 |
| | *TROLL* | 0.474 | 0.407 | 0.813 | 0.897 | 0.514 | 0.547 | 0.408 |
| RF++ | *Clip* | 0.356 | 0.251 | 0.728 | 0.897 | 0.459 | 0.538 | 0.415 |
| | *TROLL* | 0.490 | 0.355 | 0.803 | 0.891 | 0.528 | 0.563 | 0.415 |

Table 4: Success rates for individual `MATH` test datasets for Qwen2.5-7B-Instruct and Qwen3-8B models trained on `DAPO` with different advantage estimation methods. *TROLL* provides consistent benefits across methods and evaluation tasks, showing well-balanced improvements in performance. It also successfully trains GSPO without divergence, wheres *Clip* eventually causes unstable updates, as shown in Figure 8 and Figure 9.

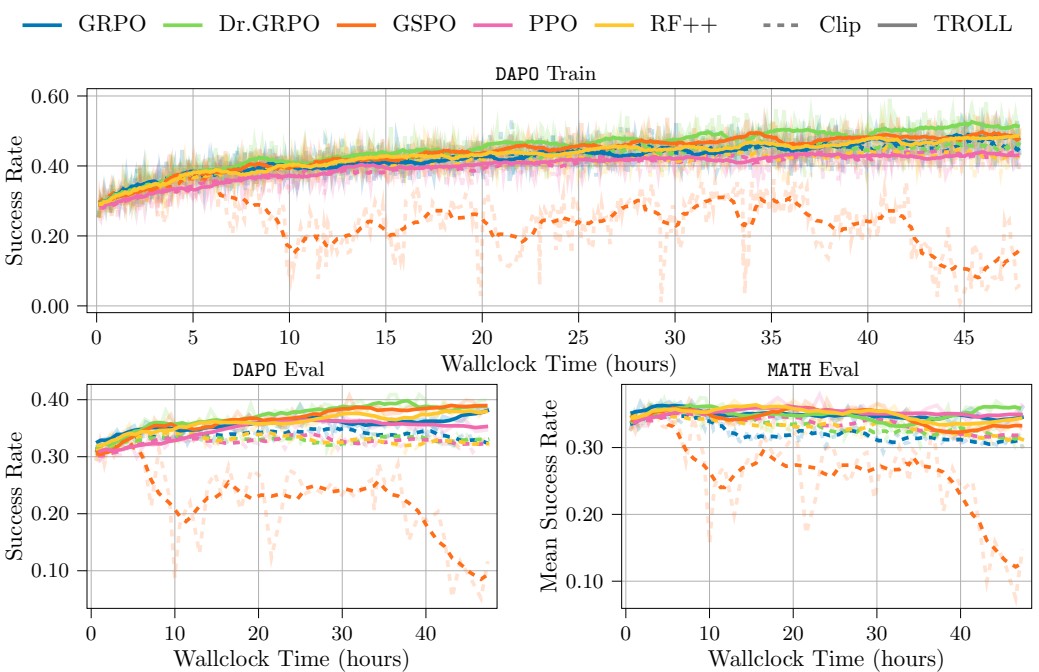

Figure 9: *TROLL* and *Clip* success rates across Qwen2.5-7B-Instruct models trained with GRPO, Dr.GRPO, GSPO, PPO and RF++ on training data (**top**), in-domain evaluation (**bottom left**) and out-of-domain evaluation (**bottom right**). Smoothed values are shown in full opacity, with original curves in the background. *TROLL* improves over the *Clip* objective for all methods. For GSPO, *Clip* eventually diverges, while *TROLL*'s optimization stays stable.

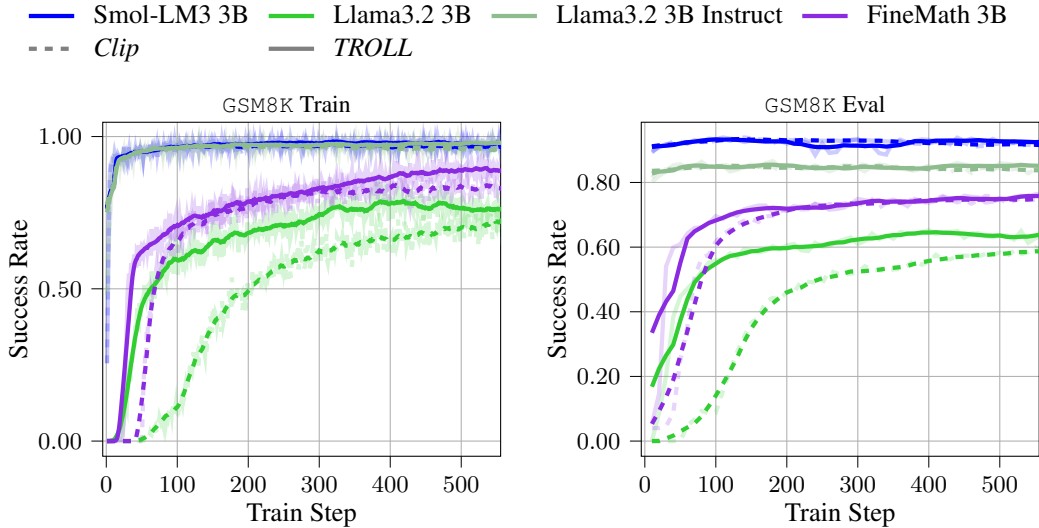

Figure 10: *TROLL* and *Clip* success rates for different 3B models trained with GRPO on the `GSM8K` training data (**left**) and evaluated on the `GSM8K` test set (**right**). Smoothed values are shown in full opacity, with original curves in the background. *TROLL* generally causes models to pick up a training signal more quickly, and exhibits more stable training behavior.

gradient method. Figure 14 finds that BAPO's adaptive clipping does not significantly improve over regular clipping in our case, and still performs worse than *TROLL*.

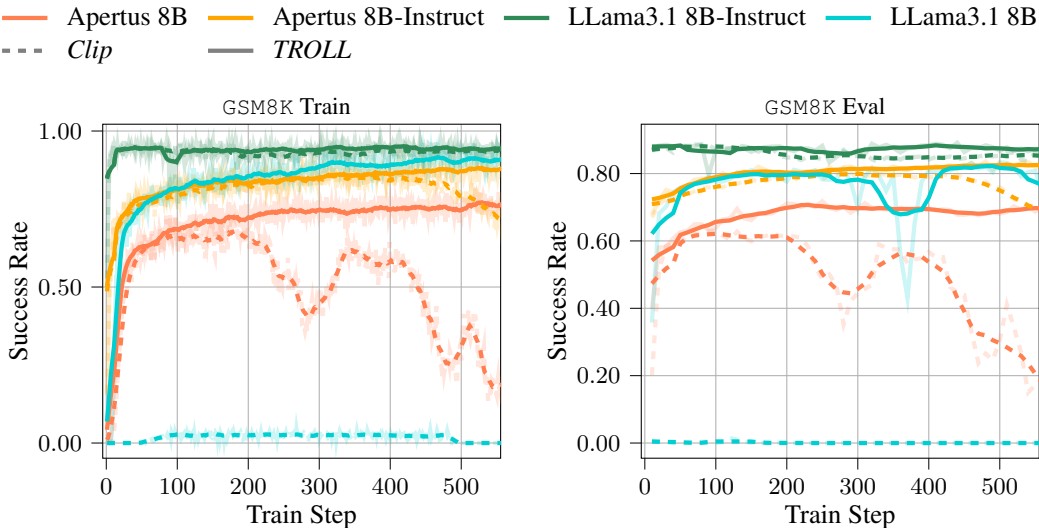

Figure 11: *TROLL* and *Clip* success rates for different 8B models trained with GRPO on the `GSM8K` training data (**left**) and evaluated on the `GSM8K` test set (**right**). Smoothed values are shown in full opacity, with original curves in the background. *TROLL* generally causes models to pick up a training signal more quickly, and exhibits more stable training behavior.

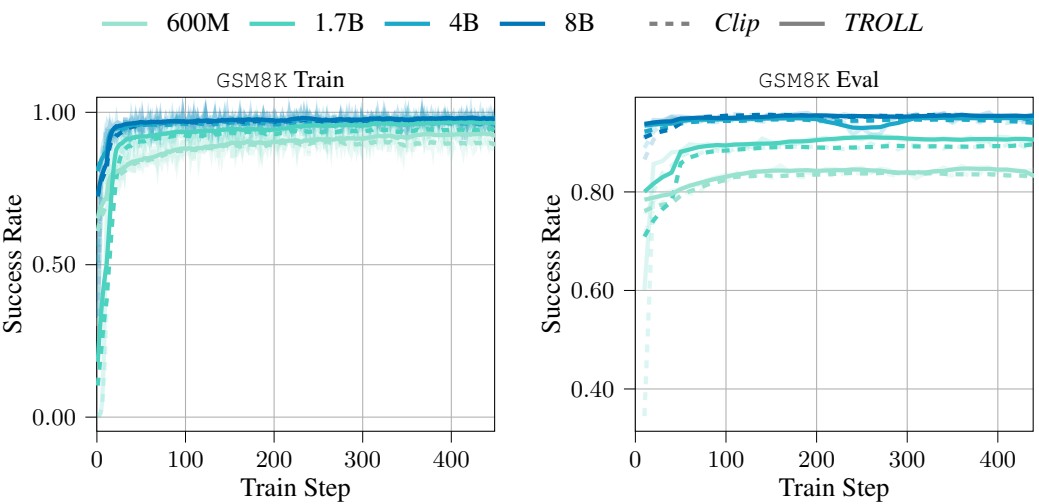

Figure 12: *TROLL* and *Clip* success rates for Qwen3 models with 600M to 8B parameters trained with GRPO on the `GSM8K` training data (**left**) and evaluated on the `GSM8K` test set (**right**). Smoothed values are shown in full opacity, with original curves in the background. Both *TROLL* and *Clip* quickly converge in all cases, although *TROLL* achieves slightly higher performance for most model sizes.

Although clipping-free GPG initially learns well, it eventually diverges during training, likely due to unstable policy updates. The figure illustrates how *TROLL* can benefit such non-clipping-based policy gradient methods. Unlike PPO-style clipping, which relies on a ratio formulation, *TROLL* limits updates by enforcing trust regions directly on the LLM output. Thus, *TROLL* can in principle be applied to any LLM post-training algorithm, including GPG. In our experiments, this approach effectively resolves GPG's stability issues, as demonstrated by the GPG(*TROLL*) variant.

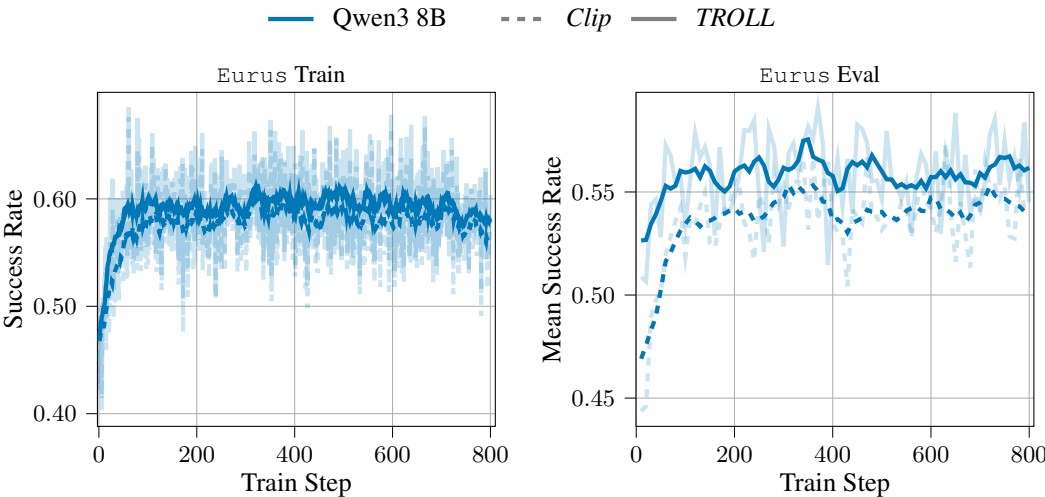

Figure 13: *TROLL* and *Clip* success rates for Qwen3-8B trained with GRPO on the `Eurus-Math` training data (**left**) and evaluated on the `Eurus-Math` test set (**right**). Smoothed values are shown in full opacity, with original curves in the background. *TROLL* performs slightly but consistently better during training, and generalizes well to the test set.

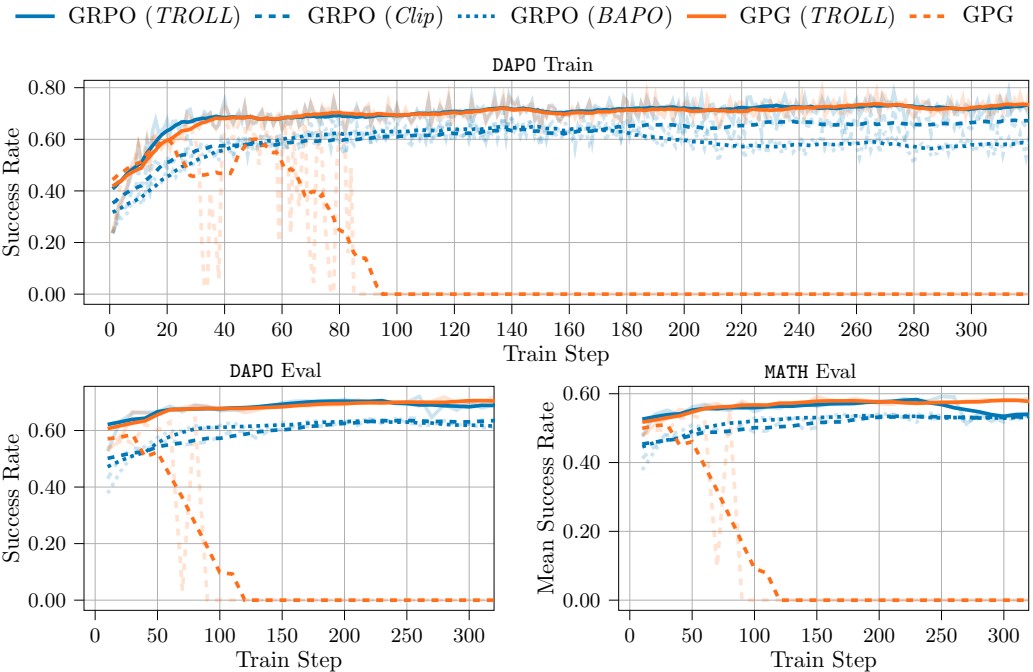

Figure 14: Success rates for Qwen3-8B on `DAPO` for training data (**top**), in-domain evaluation (**bottom left**) and out-of-domain evaluation (**bottom right**). Smoothed values are shown in full opacity, with original curves in the background. *TROLL* performs well when combined with either GRPO or the clipping-free GPG baseline, while vanilla GPG eventually diverges. BAPO's adaptive clipping slightly improves over regular *Clip* on the evaluation datasets, but both underperform *TROLL* by a fair margin.

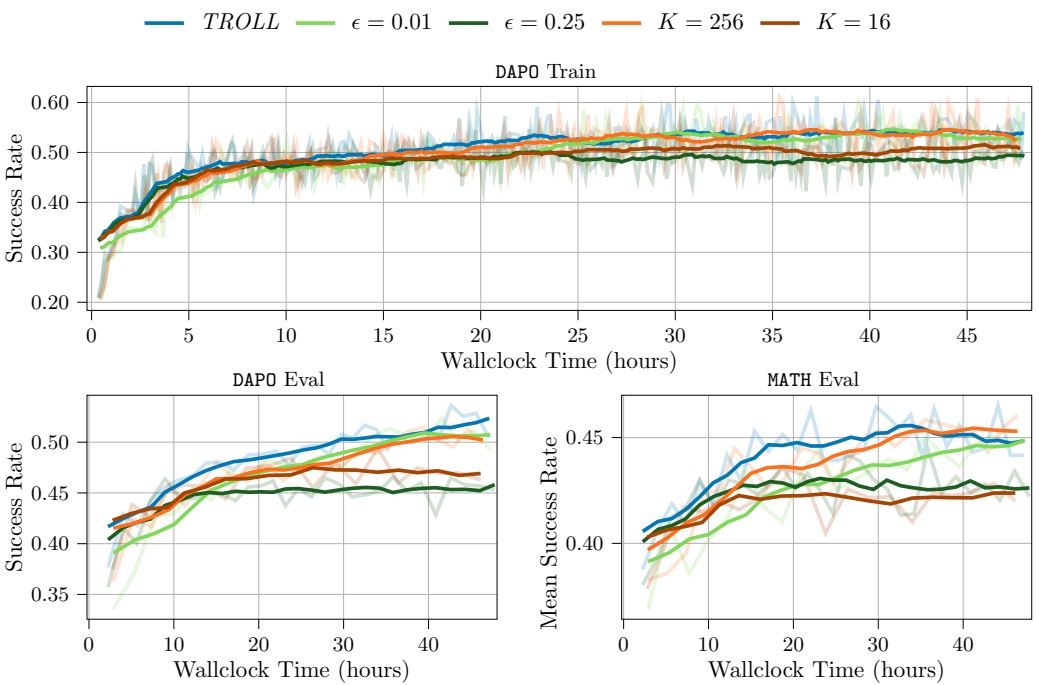

Figure 15: Qwen3-1.7B trained with GRPO using the *TROLL* projection compared to different hyperparameter choices on `DAPO` training data (**top**), in-domain evaluation (**bottom left**) and out-of-domain evaluation (**bottom right**). Smoothed values are shown in full opacity, with original curves in the background. *TROLL* works well for reasonable KL bounds $\epsilon$ and top-$K$ logit selections, but slightly degrades for too-large bounds and too few kept logits.

# E  FURTHER ANALYSIS

## E.1  PROJECTION PARAMETER SENSITIVITY

We experiment with different KL bounds, testing $\epsilon=0.01$ and $\epsilon=0.25$ instead of the default $\epsilon=0.05$. Additionally, we try different levels of sparsification. We switch the maximum number of kept tokens from $K=64$ to a lower $K=16$ and a higher $K=256$, adjusting the distribution mass threshold $\delta$ from $1e-5$ to $1e-4$ and $1e-6$ accordingly. Figure 15 shows that a lower KL bound $\epsilon$ for the projection leads to slower learning, but eventually reaches comparable performance. In contrast, a higher KL bound leads to worse performance, presumably because the policy moves too quickly during update steps. Reducing the number of kept tokens leads to worse overall performance, which is likely caused by incorrect KL estimates and thus sub-optimal projections. A higher amount of kept tokens does not yield any additional benefit, however, suggesting that $K=64$ and $\epsilon=1e$-5 maintain a sufficiently close approximation of the real policy logit distributions.

## E.2  BATCH SIZE

*TROLL*'s trust region projection promises more stable policy updates by constraining the difference between the new policy and the policy used to generate the data, in our case the responses. To evaluate this stability, we experiment with different batch sizes for Qwen3-1.7B trained with GRPO on `DAPO`, leaving all other parameters untouched. Here, doubling the batch size doubles the number of gradient steps between data collection, thus increasing the difference between the old policy $\pi_{\text{old}}(o_t \mid q, o_{<t})$ and the current policy $tarpol$ over the course of the training step. Figure 16 shows that *TROLL* remains stable when increasing the batch size from our default of $256$ to $2\,048$, indicating that *TROLL* can successfully train on older data due to its projection. In comparison, *Clip* shows a gradual degradation in performance for each doubling of the batch size, likely because the

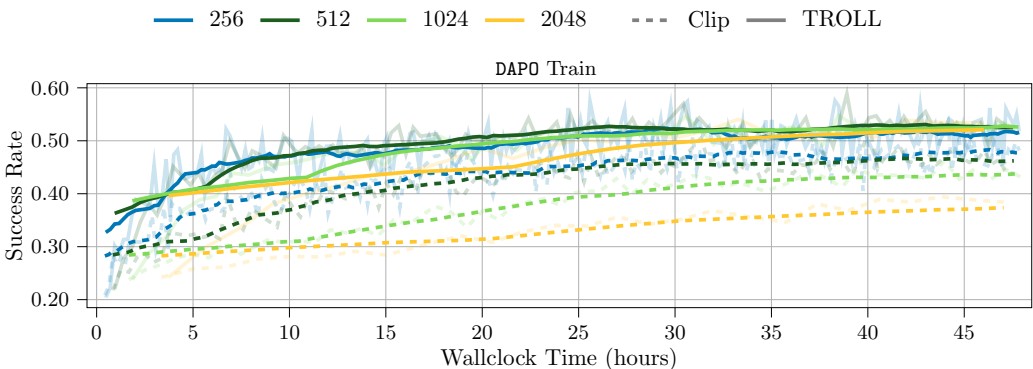

Figure 16: *TROLL* and *Clip* training success rates for Qwen3-1.7B trained with GRPO on `DAPO` for different batch sizes. *TROLL* remains stable for larger batch sizes, while *Clip* gradually and consistently degrades when increasing the batch size.

clipping does not address a potential divergence of the current policy to the policy that was used to generate the data.

### E.3 GENERATED SEQUENCE ANALYSIS

We analyze the behavior of different Qwen3 and Qwen2.5 model sizes trained on `DAPO` using GRPO in Figure 18. There is a general trend that larger models require fewer selected tokens to satisfy the sparsity mass threshold of $\delta{=}1e{-}5$, which is consistent with established LLM scaling laws (Kaplan et al., 2020). Here, as little as $5{-}10$ tokens are sufficient to capture most of the mass for the larger models. For the larger Qwen3 models, this trend appears less pronounced, likely because these models are to some extent saturating the `DAPO` benchmark. We also observe clear differences in response length dynamics over training. *TROLL* generally adapts the token length much faster than *Clip*. *TROLL* reduces the response length for Qwen3, while increasing it for Qwen2.5-Instruct. This difference originates in the different behavior of the pretrained models used to initialize learning, as the Qwen3 models tend to generate much longer responses, presumably due to their built-in thinking mode. After the RL fine-tuning with *TROLL*, the response lengths of both model families are more similar. In contrast, models trained with *Clip* show much slower shifts in response length. This quicker adjustment under *TROLL* aligns with the faster performance gains observed in both model families. Finally, both approaches clip or project slightly more than $0.1\%$ of tokens for most of the training, but *TROLL*'s projection exhibits a lot more variance and tends to increase in later training stages, potentially suggesting a more active involvement in the learning process. In some cases *TROLL* projects a lot more aggressively, although this increase in projections does not cause a degradation in model performance.

### E.4 TROLL OVERHEAD

From a practical perspective, the number of selected tokens by the sparsification and the computational and memory overhead of *TROLL* are relevant. Fair comparison of the computation and memory overhead are tricky, as both the response length and number of kept tokens varies during training and models. Initially, the average number of logits needed to achieve the desired total mass changes quickly, and small models tend to require significantly more logits due to their higher perplexity.

We therefore evaluate at the initial state by preventing updates with tight KL and clip ratio bounds. Note that setting a low, i.e. zero, learning rate is not representative since the LLM outputs must change to activate the trust region projection. Table 5 compares the memory and runtime overhead of using *TROLL* and GRPO with small Qwen3 models on `GSM8K` trained on 4x Nvidia A100-40GB GPUs. *Chunked* refers to another simple memory reduction trick, where we normalize and sparsify only a chunk, in this case size 1024, at once and avoid the dense single precision upcast of the entire mini batch. To compare the different model sizes with different response length characteristics fairly,

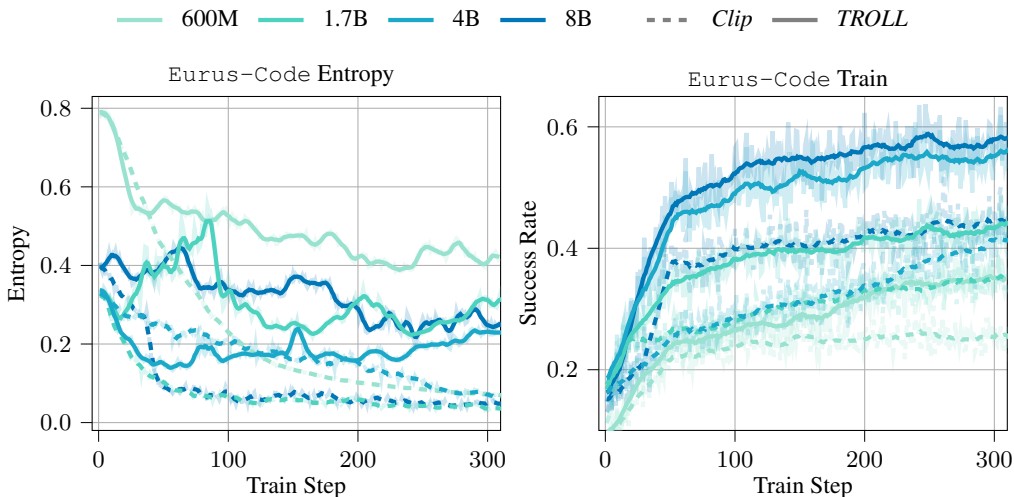

Figure 17: *TROLL* and *Clip* entropy (**left**) and success rate (**right**) for different Qwen3 models trained with GRPO on the `Eurus-Code` training data. Smoothed values are shown in full opacity, with original curves in the background. *TROLL* generally causes less decrease in token entropy while *Clip* shows a strong negative correlation between success rate and entropy. The quick improvement of Qwen3-8B *Clip* around step 40 coincides with a rapid drop in entropy.

| Metric (Method) | Qwen3-0.6B | Qwen3-1.7B | Qwen3-4B |
|---|---|---|---|
| VRAM (*Clip*) | 25.415 GiB | 28.418 GiB | 34.574 GiB |
| VRAM (*TROLL*) | 27.868 GiB | 30.663 GiB | 36.837 GiB |
| VRAM (*TROLL* Chunked) | 27.227 GiB | 29.994 GiB | 36.157 GiB |
| VRAM Delta (Chunked) | +1.812 GiB (+7.1%) | +1.576 GiB (+5.5%) | +1.583 GiB (+4.6%) |
| Runtime (*Clip*) | 30.874 s | 43.372 s | 85.133 s |
| Runtime (*TROLL*) | 46.715 s | 49.053 s | 90.570 s |
| Runtime (*TROLL* Chunked) | 47.600 s | 50.629 s | 92.906 s |
| Runtime Delta (Chunked) | +16.726 s (+54.2%) | +7.257 s (+16.7%) | +7.773 s (+9.1%) |

Table 5: Max allocated VRAM and runtime of one iteration. The smallest 0.6B models does not fully saturate the GPU, so the Delta results differ from the larger models. The projection overhead is independent of the model size and already below ten percent for the small 4B model and slower A100 GPU. The advantage of the chunked sparsification depends on the micro batch size, so the benefit is larger for bigger GPUs.

we clip all answers to just 256 tokens. Note that this length is just short enough that most answers are clipped, while some prompts are still solved. Then all models have almost the maximal 256 response tokens *on average* (about 255.4), yet still have an update gradient to reach the trust region boundary.

The responses for a single prompt with a GRPO group size of 8 and this significantly simplified sequence length of 256, float-32 representations for the probabilities, and Qwen3's tokenizer with a vocabulary of 151 936 tokens yields a memory overhead of

$$256 \cdot 8 \cdot 151936 \cdot 4B \approx 1.16\text{GiB} \tag{23}$$

for the dense implementation. Sparsification instead requires an average of $5-10$ logits per token (Figure 18 top), reducing the memory to less than $1\text{MiB}$. For each iteration, all methods need to store a rollout buffer of answers, in our case of size 256. In addition, the current mini-batch for the policy update needs to be stored. While this overhead can be reduced to a single answer with gradient accumulation, the rollout buffer still needs to store all outputs of the old policy. The total memory overhead reported in Table 5 with 256 prompts is only $\approx 6.3$ GiB across the four GPUs while just the dense distribution storage would already require $\approx 296$ GiB of memory, showing the necessity of our sparsification.

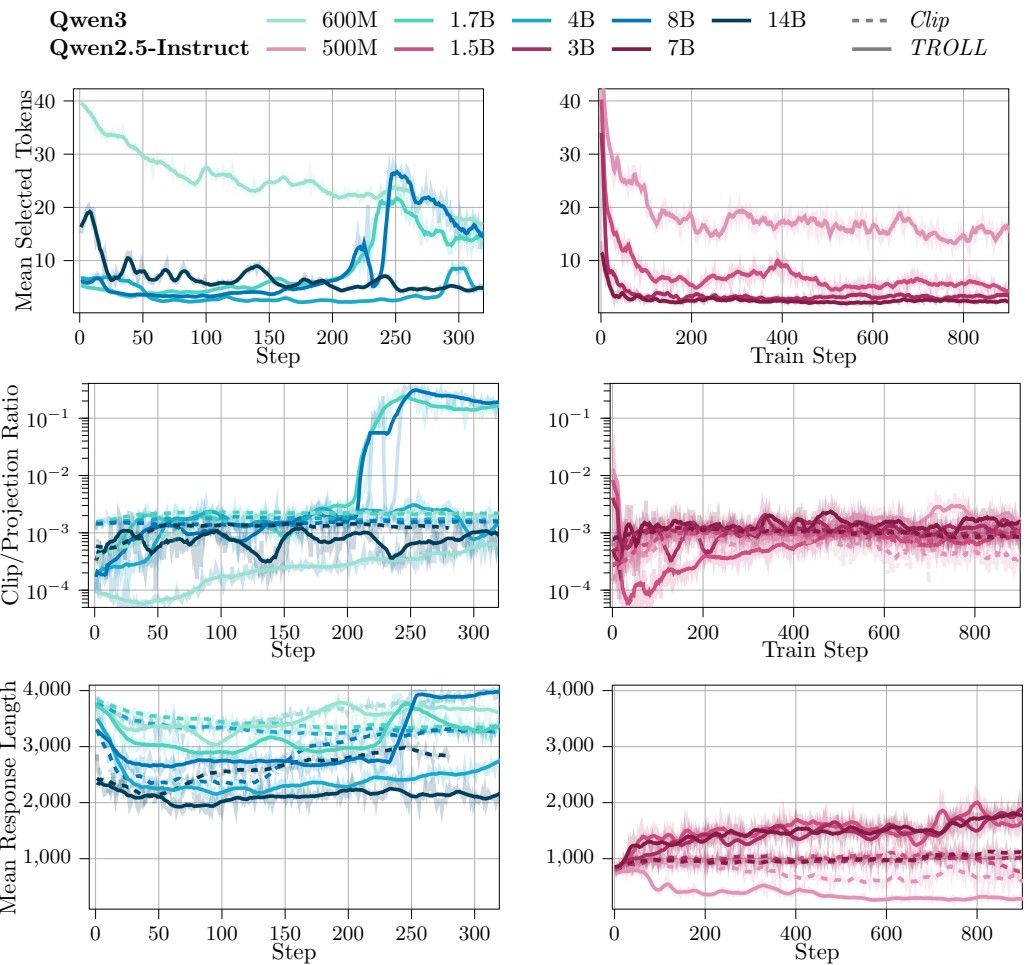

Figure 18: Training dynamics of Qwen3 models on `DAPO` using GRPO. Smoothed values are shown in full opacity, with original curves in the background. Larger models need fewer tokens to meet the sparsity threshold of $\delta = 10^{-5}$. While both approaches affect $\sim 0.1\%$ of tokens most of the time, *TROLL* tends to increase projection later during training without harming performance. *TROLL* quickly adjusts response lengths while achieving higher success rates, whereas *Clip* is slower to alter the response length over time.

## F EXAMPLE GENERATIONS

**Prompt 1**

```
1  system
2  Your task is to follow a systematic, thorough reasoning process
       before providing the final solution. This involves analyzing,
       summarizing, exploring, reassessing, and refining your thought
       process through multiple iterations. Structure your response
       into two sections: Thought and Solution. In the Thought section,
        present your reasoning using the format: "<think>
3   {thoughts} </think>
4  ". Each thought should include detailed analysis, brainstorming,
       verification, and refinement of ideas. After "</think>
5  ," in the Solution section, provide the final, logical, and
       accurate answer, clearly derived from the exploration in the
       Thought section.
6  user
7  A list of positive integers has the following properties:
8  $\bullet$ The sum of the items in the list is $30$.
9  $\bullet$ The unique mode of the list is $9$.
10 $\bullet$ The median of the list is a positive integer that does
       not appear in the list itself.
11  Find the sum of the squares of all the items in the list.
12 Present the answer in LaTeX format: \boxed{Your answer}.
13 assistant
```

**TROLL 14B Response**

<think> </think>
Thought: Okay, let's try to figure this out. The list has positive integers. Sum is 30. Unique mode is 9, so 9 appears more times than any other number, and no other number appears the same number of times as 9. The median is a positive integer not in the list. Let me think about the median.

Median is the middle value when sorted. If the number of elements is odd, median is the middle element. If even, it's the average of two middle numbers. But the median is a positive integer not in the list. So if the number of elements is odd, the median is an element of the list, which contradicts. Therefore, the number of elements must be even. Let me denote n as the number of elements, even. So n is even, say n=2k. Then median is average of k-th and (k+1)-th elements. Since median is integer, so the average of those two elements is integer, so they are both integers and their sum is even. So they are both even or both odd.

Also, mode is 9, so 9 appears at least twice. And no other number appears that many times. Let me try to find possible lists.

Let me try with n=6 elements. Then median is average of 3rd and 4th elements. Suppose elements sorted: a, b, c, d, e, f. Median is (c + d)/2, which is integer, and not in the list. So c + d is even, and (c + d)/2 is not equal to any element.

Sum a + b + c + d + e + f = 30.

Mode is 9, so 9 appears more than any other number. Let's try with two 9s. If two 9s, then other numbers can appear at most once. Let's try.

If two 9s. Let me try to have two 9s. Let me assume elements are [x, y, z, w, v, u], sorted. Suppose two 9s. Let me try positions. If two 9s, they could be in positions 1 and 2, but then median is (c + d)/2. If two 9s, maybe c and d are something else.

Wait, if there are two 9s, and others are unique. Let me try an example.

Suppose list is [1, 2, 3, 4, 9, 20]. Sum is 1+2+3+4+9+20=39 too big.

Need sum 30. Maybe more numbers. Maybe more elements. Let me try n=6.

If two 9s, then other numbers are distinct. So total numbers: 9,9,a,b,c,d. All a,b,c,d distinct and different from 9.

Sum: 9+9+a+b+c+d=18 + a + b + c + d = 30 ⟹ a + b + c + d = 12.

Also, sorted list: a, b, c, d, 9, 9? No, depends on values. Wait, sorted list would be [a, b, c, d, 9, 9] if a ≤ b ≤ c ≤ d ≤9. But median is (c + d)/2. Which is integer, and not in the list. If a, b, c, d are distinct and different from 9. Let me try numbers.

Possible a, b, c, d: positive integers, distinct, not 9, sum to 12.

Possible set: 1, 2, 3, 6 (sum 12). Check if they are distinct and not 9. Yes. Then sorted list: [1,2,3,6,9,9]. Median is (3 + 6)/2 = 4.5, not integer. Not good.

Another set: 1, 2, 4, 5 (sum 12). Sorted list: [1,2,4,5,9,9]. Median is (4 +5)/2=4.5. Still not integer.

Another set: 1, 3, 4, 4. But duplicates. Not allowed since mode is unique 9. So can't have two 4s.

Another set: 2, 2, 2, 6. Again duplicates. Not allowed.

Another set: 1, 1, 5, 5. Duplicates again.

How about three 9s? If three 9s, then other numbers can appear at most two times. Let me try n=7 elements. Wait, but median would be 4th element, which is in the list. But median is not in list. So n must be even. So n=6, 8, etc.

Try n=8 elements. Then median is average of 4th and 5th elements. Let me try with three 9s. So numbers: 9,9,9,a,b,c,d,e. Sum is 27 + a + b + c + d + e = 30 ⇒ a + b + c + d + e = 3. Since positive integers, they are 1,1,1,0,... but 0 not allowed. So impossible.

n=6 elements, three 9s: sum would be 27 + others. If n=6, three 9s, others sum to 3. Only possible if other numbers are 1,1,1. But then mode is 9 and 1, which contradicts unique mode. So invalid.

So back to n=6, two 9s. Maybe the median is integer. For example, if c and d are both even or both odd. For example, if c=3, d=5: median 4, which is not in list. If list is [1, 2, 3, 5, 9, 10]. Sum is 1+2+3+5+9+10=30. Mode is 1,2,3,5,9,10 all appear once. No mode. Not good.

Wait, need mode 9. So need more 9s. Maybe four 9s. n=7 elements: median is 4th element. Which would be 9, but median is not in list. Contradiction. n=8 elements: median is average of 4th and 5th elements. Let me try n=8. Four 9s. Others can appear at most three times. Sum: 4*9=36 >30. Too big.

n=6 elements, two 9s. Maybe other numbers have duplicates but less than two. Wait, no, if two 9s, others can appear once. So previous attempt.

Wait, what if there are two 9s, and two of another number? But mode is unique, so no. So all others appear once.

Wait, maybe n=5 elements. Median is 3rd element, which is in list. But problem says median not in list. So n must be even.

Maybe n=4 elements. Median is average of 2nd and 3rd elements. Let's try n=4. Two 9s. Sum is 9+9+a+b=18 +a +b=30 ⇒a +b=12. a and b are positive integers, sorted: a ≤ b ≤9 or something. If sorted list: [a, b, 9, 9]. Median is (b +9)/2. Must be integer, so b +9 even ⇒b odd. Also median not in list. So (b +9)/2 ≠ a, b, 9.

Let me try b=5, then median is (5+9)/2=7. 7 not in list. a=12 -5=7. So list is [7,5,9,9]. But sorted: [5,7,9,9]. Median is (7+9)/2=8, which is not in list. Okay. Mode is 9. Sum is 5+7+9+9=30. Perfect!

Wait, sum is 5+7+9+9=30. Yes. Median is 8, which is not in list. Mode is 9, unique. All conditions satisfied.

So the list is [5,7,9,9]. Sum of squares: 25 + 49 + 81 + 81 = 25 + 49 = 74, 74 + 81 = 155, 155 +81= $\boxed{236}$ .

Wait, but wait: the median is (7 +9)/2=8, which is integer and not in list. Mode is 9, which appears twice, others once. Perfect.

So answer is $\boxed{236}$ .

Wait, but let me check if there are other possible lists. For example, if a=3, b=9, but then b=9, a=3. List sorted [3,9,9,9]. Mode is 9, median is (9+9)/2=9, which is in list. Invalid.

Another case: a=5, b=7 as above. Another possibility: a=1, b=11. But b=11 >9, but sorted list [1,11,9,9] is invalid, need to sort: [1,9,9,11]. Median is (9 +9)/2=9, which is in list. Invalid.

a=3, b=9: sorted [3,9,9,9], median 9 invalid.

a=5, b=7: valid.

a=7, b=5: same as above.

a=2, b=10: sorted [2,9,9,10]. median (9+9)/2=9 invalid.

a=4, b=8: sorted [4,8,9,9]. median (8+9)/2=8.5 not integer.
a=6, b=6: sorted [6,6,9,9]. mode is 6 and 9, invalid.
Thus only possible list is [5,7,9,9], sum of squares $\boxed{236}$.
</think>
Thought: After checking all possibilities, the only valid list is [5,7,9,9], which satisfies all conditions.
Solution: The sum of the squares of all the items in the list is $\boxed{236}$.

---

**Clip 14B Response**

<think> </think>
Thought: Okay, let's see. I need to find a list of positive integers where the sum is 30, the unique mode is 9, and the median is a positive integer that doesn't appear in the list. Then, I have to find the sum of the squares of all the items.

First, let's recall what each term means. The mode is the number that appears most frequently. Since it's the unique mode, 9 must appear more times than any other number. So, there has to be at least two 9s, and no other number can appear that many times.

The median is the middle number when the list is ordered. If the list has an odd number of elements, the median is the middle one. If even, it's the average of the two middle numbers. But the problem says the median is a positive integer that doesn't appear in the list. So, if the list has an even number of elements, the average of the two middle numbers must be an integer not in the list. If odd, the middle number must not be in the list. But since the median is a positive integer, and the list contains positive integers, that's possible.

Let me try to figure out the possible lengths of the list. Let's denote the length of the list as n.

If n is odd, say n = 2k + 1, then the median is the (k+1)th element. This median must not be in the list. But if the median is not in the list, then the (k+1)th element is some number not present in the list. But all elements are positive integers, so the median must be an integer. But how can the median not be in the list? For example, if the list is [1, 2, 3, 4, 5], the median is 3, which is in the list. But if the list is [1, 2, 4, 5], the median is (2+4)/2 = 3, which is not in the list. So, for even n, the median can be an integer not in the list.

So, maybe the list has an even number of elements. Let's assume that. Let's try n = 4. Then the median is the average of the second and third elements. Let's say the list is [a, b, c, d], sorted. Then median is (b + c)/2. This must be an integer not in the list.

But also, the sum is 30. And the mode is 9. So, 9 must appear at least twice. Let's try to construct such a list.

If n = 4, and mode is 9, then at least two 9s. Let's say two 9s. Then the other two numbers must be different from each other and from 9, or one of them could be same as another but not more than once. But since mode is unique, no other number can appear twice.

So, for example, [x, 9, 9, y], sorted. Then the median is (9 + 9)/2 = 9. But 9 is in the list, which violates the condition that the median is not in the list. So that's no good.

So, if we have two 9s, and the median is not 9, then the two middle numbers must not both be 9. So, maybe the list is [a, 9, 9, b], but sorted. If a and b are such that the median is not 9. But if a <= 9 <= b, then the median is (9 + 9)/2 = 9. So that's not possible. So, maybe we need more than two 9s?

If we have three 9s. Let's try n = 5. Then the median is the third element. If the list is [a, b, 9, c, d], sorted. Then the median is 9. But 9 is in the list, which violates the condition. So that's no good.

If n = 6. Then the median is the average of the third and fourth elements. Let's say the list is [a, b, x, y, c, d], sorted. Then median is (x + y)/2. This must be an integer not in the list.

Also, mode is 9. So, 9 must appear more times than any other number. Let's say 9 appears three times. Then other numbers can appear at most two times.

Let me try to construct such a list. Let's say the list has three 9s. Let's arrange them in the list. For example, [a, b, 9, 9, 9, d], sorted. Then the third and fourth elements are 9 and 9. So the median is (9 + 9)/2 = 9, which is in the list. Not allowed.

So, maybe the three 9s are not in the middle. Let's say the list is [a, 9, 9, 9, b, c], sorted. Then the third and fourth elements are 9 and 9. Again, median is 9. Not allowed.

So, maybe the three 9s are spread out. For example, [a, 9, b, 9, c, 9], sorted. Then the third and fourth elements are b and 9. So the median is (b + 9)/2. This must be an integer not in the list.

Let me pick b such that (b + 9) is even. So, b must be odd. Let's say b = 7. Then median is (7 + 9)/2 = 8. So 8 is the median, and it's not in the list. That works.

So, the list is [a, 9, 7, 9, c, 9], sorted. Wait, but sorted order would be [a, 7, 9, 9, c, 9]. But that's not sorted unless a <= 7 and c >= 9.

Let me make it sorted. Let's say a <= 7 <= 9 <= c <= 9. So, a <= 7, and c >= 9. But since there are three 9s, maybe c is 9. So the list is [a, 7, 9, 9, 9, 9]. But then the third and fourth elements are 9 and 9. Median is 9, which is in the list. Not allowed.

Hmm. So, maybe the three 9s are not consecutive. Let me try [a, 7, 9, 9, c, d], sorted. Then the third and fourth elements are 9 and 9. Median is 9. Not allowed.

So, maybe we need four 9s. Let's try that. If there are four 9s, then the mode is 9, and no other number appears more than three times.

Let's try n = 7. Then the median is the fourth element. If the list is [a, b, c, 9, d, e, f], sorted. Then the median is 9, which is in the list. Not allowed.

If n = 8. Then the median is the average of the fourth and fifth elements. Let's say the list is [a, b, c, x, y, z, w, v], sorted. Then median is (x + y)/2. This must be an integer not in the list.

If we have four 9s. Let's say the list is [a, b, c, 9, 9, 9, 9, d], sorted. Then the fourth and fifth elements are 9 and 9. Median is 9, which is in the list. Not allowed.

So, maybe the four 9s are spread out. For example, [a, b, 9, c, d, 9, e, 9], sorted. Then the fourth and fifth elements are c and d. So, median is (c + d)/2. Let's pick c and d such that this is an integer not in the list.

Let me try c = 7 and d = 9. Then median is (7 + 9)/2 = 8. Not in the list. So that works.

So the list is [a, b, 9, 7, 9, 9, e, 9], sorted. Wait, but sorted order would be [a, b, 7, 9, 9, 9, e, 9]. So, a and b must be <= 7, and e >= 9.

Let me pick a and b as small as possible. Let's say a = 1, b = 2. Then the list is [1, 2, 7, 9, 9, 9, e, 9]. Sorted, it would be [1, 2, 7, 9, 9, 9, e, 9]. But e must be >= 9. Let's say e = 9. Then the list is [1, 2, 7, 9, 9, 9, 9, 9]. But then the fourth and fifth elements are 9 and 9. Median is 9, which is in the list. Not allowed.

So, e must be greater than 9. Let's say e = 10. Then the list is [1, 2, 7, 9, 9, 9, 10, 9]. Sorted: [1, 2, 7, 9, 9, 9, 9, 10]. Now, the fourth and fifth elements are 9 and 9. Median is 9, which is in the list. Not allowed.

So, maybe c and d are not 7 and 9. Let's try c = 7 and d = 10. Then median is (7 + 10)/2 = 8.5, not an integer. Not allowed.

c = 6 and d = 10. Median is (6 + 10)/2 = 8. Good. So, the list is [a, b, 6, 9, 10, 9, e, 9], sorted. Wait, sorted would be [a, b, 6, 9, 9, 9, e, 10]. So, a and b <= 6, e >= 9.

Let me pick a = 1, b = 2. Then the list is [1, 2, 6, 9, 9, 9, e, 10]. Sorted: [1, 2, 6, 9, 9, 9, e, 10]. The fourth and fifth elements are 9 and 9. Median is 9, which is in the list. Not allowed.

So, maybe the fourth and fifth elements are not both 9. Let me try to have the four 9s spread out more. For example, [a, b, 9, c, d, 9, e, 9], sorted. Then the fourth and fifth elements are c and d. Let me pick c = 7 and d = 8. Then median is (7 + 8)/2 = 7.5, not integer.

c = 7 and d = 9. Median is 8. So, the list is [a, b, 9, 7, 9, 9, e, 9], sorted: [a, b, 7, 9, 9, 9, e, 9]. Again, fourth and fifth are 9 and 9. Median is 9.

This seems tricky. Maybe I need to have more than four 9s? Let's try five 9s.

If n = 9. Then the median is the fifth element. If the list is [a, b, c, d, 9, e, f, g, h], sorted. Then the median is 9, which is in the list. Not allowed.

If n = 10. Then the median is the average of the fifth and sixth elements. Let's say the list has five 9s. So, [a, b, c, d, 9, 9, 9, 9, 9, e], sorted. Then the fifth and sixth elements are 9 and 9. Median is 9, which is in the list. Not allowed.

So, maybe the five 9s are spread out. For example, [a, b, c, 9, d, e, f, 9, g, 9], sorted. Then the fifth and sixth elements are d and e. Let me pick d = 7 and e = 8. Then median is (7 + 8)/2 = 7.5, not integer.

d = 7 and e = 9. Median is (7 + 9)/2 = 8. Not in the list. So that works.

So the list is [a, b, c, 9, 7, 9, f, 9, g, 9], sorted. Wait, sorted would be [a, b, c, 7, 9, 9, f, 9, g, 9]. So, a, b, c <= 7, f >= 9, g >= 9.

Let me pick a = 1, b = 2, c = 3. Then the list is [1, 2, 3, 7, 9, 9, f, 9, g, 9]. Sorted: [1, 2, 3, 7, 9, 9, f, 9, g, 9]. The fifth and sixth elements are 9 and 9. Median is 9, which is in the list. Not allowed.

So, maybe the fifth and sixth elements are not both 9. Let me try to have the five 9s spread out more. For example, [a, b, 9, c, d, e, f, 9, g, 9], sorted. Then the fifth and sixth elements are d and e. Let me pick d = 7 and e = 8. Then median is (7 + 8)/2 = 7.5, not integer.

d = 7 and e = 9. Median is 8. So, the list is [a, b, 9, c, 7, 9, f, 9, g, 9], sorted: [a, b, 7, 9, 9, 9, f, 9, g, 9]. The fifth and sixth elements are 9 and 9. Median is 9.

This is really challenging. Maybe I need to go back to n = 6 and try again.

Let me try n = 6, with three 9s. The median is (third + fourth)/2. Let me have the list [a, b, 7, 9, c, d], sorted. Then the third and fourth elements are 7 and 9. Median is (7 + 9)/2 = 8. Not in the list. That works!

So, the list is [a, b, 7, 9, c, d], sorted. With three 9s. So, one of a, b, c, d is 9. Let's say c and d are 9. So, the list is [a, b, 7, 9, 9, 9], sorted. Then a and b <= 7.

Let me pick a = 1, b = 2. Then the list is [1, 2, 7, 9, 9, 9]. Sum is 1 + 2 + 7 + 9 + 9 + 9 = 37. Too much.

We need sum to be 30. So, let's adjust. Let me try a = 1, b = 1. Then the list is [1, 1, 7, 9, 9, 9]. Sum is 1 + 1 + 7 + 9 + 9 + 9 = 36. Still too much.

a = 1, b = 2, and change 7 to something smaller. But 7 is the third element. If I make it 6, then the list is [1, 2, 6, 9, 9, 9]. Sum is 1 + 2 + 6 + 9 + 9 + 9 = 36. Still too much.

Maybe a = 1, b = 1, and third element is 5. Then list is [1, 1, 5, 9, 9, 9]. Sum is 1 + 1 + 5 + 9 + 9 + 9 = 34. Still too much.

a = 1, b = 1, third = 4. Sum is 1 + 1 + 4 + 9 + 9 + 9 = 33.
a = 1, b = 1, third = 3. Sum is 1 + 1 + 3 + 9 + 9 + 9 = 32.
a = 1, b = 1, third = 2. Sum is 1 + 1 + 2 + 9 + 9 + 9 = 31.
a = 1, b = 1, third = 1. Sum is 1 + 1 + 1 + 9 + 9 + 9 = 30. Perfect!

So the list is [1, 1, 1, 9, 9, 9]. Sorted. Let's check the conditions:

- Sum is 1 + 1 + 1 + 9 + 9 + 9 = 30. Check. - Mode is 9, which appears three times. Other numbers appear once or twice. So unique mode is 9. Check. - Median is (third + fourth)/2 = (1 + 9)/2 = 10/2 = 5. 5 is not in the list. Check.

So this list satisfies all conditions!

Now, the sum of the squares is $1^2 + 1^2 + 1^2 + 9^2 + 9^2 + 9^2 = 1 + 1 + 1 + 81 + 81 + 81 = 3 + 243 = 246$.

So the answer is $\boxed{246}$.

But wait, let me check if there are other possible lists. For example, could there be a list with more elements?

But given the constraints, this seems to be the only possible list. So I think

