# OpenReview forum: "TROLL: Trust Regions Improve Reinforcement Learning for Large Language Models"
_ICLR.cc/2026/Conference — ICLR 2026 Oral_

### Official Review · Reviewer_RHmH · 2025-10-17

**Soundness:** 2
**Presentation:** 3
**Contribution:** 2
**Rating:** 6
**Confidence:** 4

**Summary:**

In this paper, authors considers PPO-like approaches, more specifically approaches that make use of a clipped importance sampling ratio. Instead of clipping this ratio, they propose to use a projection that keeps the KL within a certain range. In practice, this is an extension of Otto et al (2021) from Gaussian distributions to categorical distributions, with an additional spare and efficient representation of token distributions to cope with the usually huge vocabulary / action space of LLMs. The proposed approach is quite general (in replacing clipping by this projection for any methods making use of this clipping). A quite thorough experimental study is provided, showing the advantage of the approach compared to clipping, and providing interesting ablations, including illustrating the small computational overhead.

**Strengths:**

* The paper is overall well written and structured
* Clipping is a commonly used heuristic, and any approach improving it can have a quite wide impact. This approach in particular is well motivated (clipping is indeed a heuristic approx of trust region) and seems to be quite efficient, with specific care to specificities of LLMs (like huge action space)
* The experimental study is quite thorough, provide interesting ablations, and showcase the advantage of the proposed approach in many cases.

**Weaknesses:**

* There are clarity issues about the proposed approach, notably what optimization problem is really solved, how the trust-region projection interplays with the more general RL problem, or what justifies the heuristic of Eq (5). See questions below for more details.
* Given that the paper studies an alternative to clipping, that’s a bit a pity that the authors do not consider as related works or even possibly baselines other alternatives to clipping, including approaches that even avoid relying on importance sampling. See questions below for more details and specific references.

**Questions:**

### Clarity

* Maybe nitpicking, but the paper talks about on-policy approaches all along. If it was truly on-policy, there would be no importance sampling and then no clipping.
* It is unclear what is the overall optimization problem. It seems that problem (1) (RL objective) is solved with problem (3) (trust region projection), but how do they interplay? The term $\tilde{\pi}$ is the optimization variable of Eq (1) (as a side note, it is much clearer to write $J_{ratio}(\tilde{\pi})$ rather than just $J_{ratio}$, it is a function, making the argument explicit helps). Let call $P(\tilde{\pi})$ the result of Eq (3) (the argmin). Do we solve for $J_{ratio}(P(\tilde{\pi}))$ or something else? How does it relate to the classic KL constrained RL problem (eg as presented in TRPO)?
* In Eq (5), it is not clear what gradient clipping means in this context, how it can be applied (and why) only partly, or what is the overall justification for doing Eq (5), and its implications. It is not even clear what policy is optimized there. Overall, this should be really clarified.
* What is also unclear is towards what we regularize (what is $\pi_{old}$). We can regularize towards two policies, the initial one, and the previous one (in a policy iteration context, what is fundamentally PPO or TRPO). Often, regularization is made towards both, through different term (eg, GRPO consider regularization towards both, the previous one through importance ration and clipping and the inital policy through the regularization term). Please clarify what $\pi_{old}$ is in your case.

### Related works and baselines

Given that the paper addresses the limitations of clipping, it misses a few approaches from the literature, in the context of LLMs, with the same or a very similar objective, either proposing an alternative to clipping the importance sampling ratio, or even removing the importance sampling ratio. Notably, the following papers could be discussed, and possibly considered as baselines:
* [A] generalizes clipping by clipping the importance ratio within the gradient rather than within the objective, asymmetrically
* [B] is a policy-gradient approach (in the sense of optimizing for a policy), relying on an additional value network but without any importance sampling, while being off-policy
* [C,D] is an off-policy policy gradient without importance sampling
* [E] is a method relying rather on Q-functions, but as so being off-policy without importance sampling (see also references therein for older approaches)

[A] Tapered Off-Policy REINFORCE: Stable and efficient reinforcement learning for LLMs, Le Roux et al, 2025.
[B] Offline Regularised Reinforcement Learning for Large Language Models Alignment, Richmond et al, 2024.
[C] Contrastive Policy Gradient: Aligning LLMs on sequence-level scores in a supervised-friendly fashion, Flet-Berliac et al, 2024.
[D] Command A: An Enterprise-Ready Large Language Model, Cohere, 2025.
[E] ShiQ: Bringing back Bellman to LLMs, Clavier et al, 2025.


### Misc
* Eq (4) there’s probably a $\log$ missing around $\tilde{\pi}$

---

> ### Author Response · Authors · 2025-11-20
> **Response to Reviewer RHmH**
>
> We are grateful for the reviewer’s helpful comments and suggestions, in particular for highlighting the importance of clarifying our notation and disambiguating the use of on- vs off-policy RL. In the following, we will address the individual points mentioned.
>
> > *There are clarity issues about [...] what optimization problem is really solved, how the trust-region projection interplays with the more general RL problem, or what justifies the heuristic of Eq (5).*
>
> We thank the reviewer for this important insight. We agree that the method would benefit from additional clarification and will add pseudocode and a schematic on gradient propagation. We will also add explicit function arguments for, e.g., $J_{\theta}$ for clarity. We provide more detailed answers for the other clarity-related questions below.
>
> > *[...] the authors do not consider as related works or even possibly baselines other alternatives to clipping [...] either proposing an alternative to clipping the importance sampling ratio, or even removing the importance sampling ratio.*
>
> We agree that a more thorough discussion of alternative approaches related to improving the clipping objective is very valuable to contextualize TROLL. We will add an extended discussion on the suggested methods in the related work section. We are currently running additional experiments with GPG [1] as a clipping-free baseline and will provide a detailed setup and results in the revision.
>
> > *[...] the paper talks about on-policy approaches all along. If it was truly on-policy, there would be no importance sampling and then no clipping.*
>
> Our submission follows common RL naming conventions, which often classify PPO as an on-policy algorithm [2, 3]. We agree that this nomenclature can be confusing in the context of LLM post-training, and will clarify the relationship in the revision.
>
>
> > *It seems that problem (1) (RL objective) is solved with problem (3) (trust region projection), but how do they interplay? [...] In Eq (5), it is not clear what gradient clipping means in this context [...]*
>
> The trust region projection (Eqs. 3, 4) projects the outputs of the new policy onto a KL-based trust region around the old policy. This idea follows the general trust region paradigm [4], essentially ensuring that the new policy’s output stays sufficiently close to that of the old policy for any given update step. Once obtained, the general policy gradient objective (Eq. 1) is applied to the projected policy, as realised in Eq. 5. This equation also adds a KL regression loss between the projected and unprojected updated policy, which ensures that the unprojected policy regresses to its projection. We will clarify these relationships in the revision.
> Here, a gradient stop (or detach operation) is applied to ensure that the unprojected policy regresses to its projection, and not vice versa. Thus, the second part of Eq. 5 encourages the actual updated LLM policy to stay close to where it would be projected based on the old policy, which effectively limits its update size. We agree that the term “clipping” is ambiguous here and will make the statement more precise.
>
> > *What is also unclear is towards what we regularize*
>
> We appreciate the reviewer’s attention to detail and are happy to clarify this question. Currently, TROLL always uses the previous policy, i.e., the policy at the start of a training iteration, as its reference. We briefly mention this in ll. 186-187 below Eq. 1 and will highlight the distinction to the initial policy in the revision. Here, both TROLL's objective and clipping thus simplify to the standard policy gradient when taking a single gradient step after collecting data.
> Regularizing towards the initial, pre-trained policy is a straightforward extension that can be realized with a similar projection and regression as in Eq. 5, using something like $\pi_{\text{init}}$ instead of $\pi_{\text{old}}$. We omit this regularization as it is not necessary for the logical reasoning tasks that we consider [5], but believe it to be an interesting addition for other tasks, such as aligning to human preferences.
>
>
> > *Eq (4) there’s probably a $\log$ missing around $\tilde{\pi}$*
>
> We thank the reviewer for catching this mistake and will correct it in the revision.
>
> We appreciate the reviewer’s valuable insights and positive assessment. We are eager to engage in further discussion during the rebuttal.
>
> [1] Chu, Xiangxiang, et al. "Gpg: A simple and strong reinforcement learning baseline for model reasoning." *arXiv* (2025)
>
> [2] OpenAI. "Spinning Up in Deep RL: Proximal Policy Optimization." https://spinningup.openai.com/en/latest/algorithms/ppo.html, (accessed Nov. 2025)
>
> [3] Andrychowicz, Marcin, et al. "What matters for on-policy deep actor-critic methods? a large-scale study." *ICLR* (2021)
>
> [4] Schulman, John, et al. "Trust region policy optimization." *ICLR* (2015)
>
> [5] Yu, Qiying, et al. "Dapo: An open-source llm reinforcement learning system at scale." *arXiv* (2025).

---

> > ### Comment · Reviewer_RHmH · 2025-11-26
> > **Rebuttal acknowledgement**
> >
> > I thank the authors for their answer, which clarified how the overall projection approach works.
> >
> > I appreciate the updated discussion on alternatives to clipping or no clipping approaches, as well as authors committing to provide additional experiments, I look forward to them (as well possibly as to a short discussion on why GPG as a representative baseline).

---

> > > ### Author Response · Authors · 2025-11-27
> > >
> > > We thank the reviewer for the continued discussion, and for acknowledging the now-clarified method section.
> > >
> > > We used GPG as a representative non-clipping baseline due to its simplicity and recent empirical success. Further, GPG is readily available in the verl repository, which we use as a basis for our code. This availability ensures consistency with the other experiments given the rebuttal's limited time period.
> > > Our second revision now includes new experimental results. We show that GPG is compatible with TROLL, i.e., that its policy can be projected in the same way as those of our other experiments. This combination stabilizes and improves GPG training, achieving parity with, e.g., GRPO.
> > >
> > > We are happy to answer further questions and address remaining concerns.

---

### Official Review · Reviewer_YCtx · 2025-10-27

**Soundness:** 3
**Presentation:** 3
**Contribution:** 3
**Rating:** 6
**Confidence:** 3

**Summary:**

The paper presents a way of fine-tuning LLM with RL that uses a more principled approach to ensure the network updates are within a trust reagion by using a token-wise kl constraint objective. This is different from the standard way of fine-tuning LLM's with algorithms like MPO that use clipping. The trust reagion is also differentuable which is important for updates. The paper also presents a way of sparcificaition for scalability where only a subset of tokes is used to compute the trust region.

**Strengths:**

The paper and the algorithm are presented well and intuitively. TROLL does outperform the baseline even if only by a small margin, nevertheless I think there is a big value for a more principled approach than clipping.

**Weaknesses:**

The gain in performance is not super big, also the scale of evaluation is a bit limited as only smaller models have been tested so it is unclear how well this will scale. Additionally was seems a bit narrow is the choice of task domains that are heavily focused on mathematical reasoning tasks. TROLL should be useful in a variety of tasks and not only for one specific domain.

**Questions:**

How would this scale to bigger models?
can you provide evaluations on other tasks than mathematical reasoning to show that TROLL is generally outperforming the baselines?

---

> ### Author Response · Authors · 2025-11-20
> **Response to Reviewer YCtx**
>
> We thank the reviewer for their helpful feedback, and in particular for appreciating the presentation and theoretical foundation of the presented approach. In the following, we aim to address the individual concerns raised.
>
> > *The gain in performance is not super big [...]*
>
> We find that TROLL substantially improves over clipping in most experimental settings. Evaluating Qwen3-8B and Qwen2.5-7B in Table 1 shows that most combinations improve by 3-10% absolute, corresponding to roughly 5-15% relative improvements. Considering different models, Figure 4 and Figures 11, 12 in the Appendix indicate that TROLL proves far more robust in scenarios where clipping fails. For base models like Llama3.1-8B and Apertus-8B (Figure 4), or when using the relatively unstable GSPO (Table 1), clipping either does not learn or diverges at some point during training. Here, TROLL still works effectively, qualitatively enabling training in these settings.
> We are currently running additional experiments on different advantage estimation methods within the scope of the rebuttal. Our preliminary results indicate that here, too, using TROLL has a consistent, positive impact and is more important than the concrete choice of advantage estimation. We will add these results to the revision.
>
> > *[...] the scale of evaluation is a bit limited as only smaller models have been tested so it is unclear how well this will scale. [...] How would this scale to bigger models?*
>
> We experiment with TROLL for models from 0.5B up to 14B parameters, finding consistent improvements over clipping across these ranges. Judging by these results, as well as general [1] and RLVR-specific scaling trends [2, 3], we strongly expect TROLL to provide a similar benefit for larger models. Still, we agree that it would be interesting and valuable to scale TROLL to 100B+ models in future work, but unfortunately do not have access to sufficient hardware for these types of experiments within the rebuttal period.
>
> > *[The] task domains [...] are heavily focused on mathematical reasoning tasks. [...] can you provide evaluations on other tasks than mathematical reasoning to show that TROLL is generally outperforming the baselines?*
>
> We fully agree with the reviewer that an additional application domain would give a more complete overview of TROLL’s capabilities. We are currently running code generation experiments, where preliminary results show that TROLL improves over clipping in a similar way to the math benchmarks considered in the original submission. We are happy to provide experimental details and results in the revision once all runs are finished.
>
>
> We want to thank the reviewer again for their interesting suggestions and insights, and are happy to answer additional questions during the rebuttal.
>
> [1] Kaplan, Jared, et al. "Scaling laws for neural language models." *arXiv* (2020)
>
> [2] Touvron, Hugo, et al. "Llama 2: Open foundation and fine-tuned chat models." *arXiv* (2023)
>
> [3] Rafailov, Rafael, et al. "Direct preference optimization: Your language model is secretly a reward model." *NeurIPS* (2023)

---

> > ### Comment · Reviewer_YCtx · 2025-11-27
> >
> > Thank you for the rebuttal and for addressing my questions. I have updated my score

---

> > > ### Author Response · Authors · 2025-11-27
> > >
> > > We thank the reviewer for their positive assessment. We are happy to engage in further discussion if any questions or issues arise.

---

### Official Review · Reviewer_cNjy · 2025-11-01

**Soundness:** 2
**Presentation:** 2
**Contribution:** 3
**Rating:** 4
**Confidence:** 3

**Summary:**

Existing reinforcement learning (RL) algorithms for large language models (LLMs) including PPO and GRPO rely on clipping on policy ratios. While theoretically motivated by trust region optimization, clipping is only a crude approximation of the trust region. To this end, the authors propose TROLL, a differentiable trust region projection approach that directly enforces token-level KL constraints between discrete distributions. TROLL can directly replace the clipping objectives in PPO-like algorithms. Notably, TROLL projects the output distribution of the new, updated policy onto a KL-trust region around the old policy, and the projection can be computed in closed form.  To implement TROLL in practice, the authors introduce a sparsification scheme that discards the vast majority of effectively irrelevant, low-probability tokens. The authors verify the effectiveness of TROLL in RLVR settings on multiple LLM families across math datasets.

**Strengths:**

- This work addresses an important challenge for LLM post-training.
- The proposed method TROLL is novel.
- The authors compare multiple LLM families with various sizes in their experiments.

**Weaknesses:**

## Presentation
I personally find Section 3 (the most important algorithm section) to be poorly written and organized. In particular, I recommend the authors to include a pseudocode for the complete algorithm, from computation of $\Tilde{\pi}$, to its projection, to the calculation of final loss. Then write a separate subsection about the how the gradients are propagated and move the key results to the main text. As of now, it is difficult to understand both the forward and backward pass of the TROLL algorithm. This is the most critical concern to me. And I will update the evaluation if the presentation problem can be sufficiently addressed.

## Related Work
As PPO is one of the most classical algorithms for RL, there have been many attempts to enhance its clipping objective, e.g.,
- with adaptive clipping [1],
- with KL-regularized surrogates [3].

In particular, I am wondering why the following works of PPO (although not designed for LLMs) does not address the clipping problem of PPO. In other words, what are the unique challenges about LLMs that TROLL addresses and these prior RL works cannot.

In addition, I think [2] addresses the same challenge and the authors should consider comparing it.

[1] Trust Region-Guided Proximal Policy Optimization.

[2] BAPO: Stabilizing Off-Policy Reinforcement Learning for LLMs via Balanced Policy Optimization with Adaptive Clipping.

[3] V-MPO: ON-POLICY MAXIMUM A POSTERIORI POLICY OPTIMIZATION FOR DISCRETE AND CONTINUOUS CONTROL


## Experiments
**Marginal Performance Gain.** Despite of the complex algorithm, the performance increase on the most popular combination for the math benchmark (Qwen + GRPO) is marginal.

**Only Math Benchmark.** It would be beneficial if the authors can include more diverse RLVR benchmarks other than the math benchmarks. While it is appreciated that the authors conduct experiments on extensive benchmarks, all of them are math benchmarks and this can lead to questions about TROLL's scalability to other types of post-training tasks.

**Questions:**

- Why the authors claim that TROLL is for LLMs only? TROLL is an attempt to improve PPO, which is a general RL algorithm. Hence, TROLL should also be effective for standard (non-LLM) RL tasks.
- If I understand correctly, it seems that the dual step size $\eta^\star$ needs to be calculated for every token?
- Why the gain for GSPO much larger than GRPO and DrGRPO?
- Could the authors please explain in more details why TROLL is much more complex than

---

> ### Author Response · Authors · 2025-11-20
> **Response to Reviewer cNjy**
>
> We thank the reviewer for their constructive review, and especially for helpful suggestions to improve the method section’s presentation and extend the related work. In the following, we aim to address the individual concerns raised.
>
> > *[Section 3 should] include a pseudocode [and have] a separate subsection about how the gradients are propagated [...]*
>
> We fully agree with the reviewer that the method section can be clarified, and we are happy to add pseudocode and a small schematic on gradient propagation. We will add both to the revision and encourage the reviewer to reach out for further clarifications.
>
> > *[...] there have been many attempts to enhance [the PPO] clipping objective [...] what are the unique challenges about LLMs that TROLL addresses and these prior RL works cannot. [...]*
>
> We thank the reviewer for this critical question and for suggesting additional related work. TROLL has two unique advantages compared to clipping-based methods. First, the projection is exact, whereas clipping is known to be a crude approximation that comes with several issues [1,2]. Second, TROLL’s trust regions are fully differentiable, yielding training feedback for cases where clipping simply truncates the gradient. Both advantages are crucial for LLM training as they represent a more faithful implementation of the trust region paradigm for RL post-training.
>
> V-MPO [3] regularizes the expected KL, while TROLL operates on a per-token leve. Since LLM post-training focuses on optimizing a few crucial tokens, averaging via an expected KL would lack the resolution to provide a meaningful regularization in this setting. We will discuss this concept and its related work in more detail in the revision.
>
>
> > *[BAPO] addresses the same challenge and the authors should consider comparing it.*
>
> While BAPO [4] has only been uploaded to arXiv after our initial submission, we agree that it is highly relevant for TROLL. We are currently running additional experiments on it and are happy to provide results in the revision. We see such adaptive approaches as orthogonal to TROLL, and think that, e.g., adaptively scaling TROLL’s trust region could be an interesting direction for future work.
>
> > *Despite of the complex algorithm, the performance increase [...] is marginal.*
>
> Once implemented, TROLL acts as a mostly drop-in replacement of clipping. We strongly believe that it substantially improves over clipping and more than justifies its additional cost. We refer to the general answer for a more detailed discussion and are happy to go into more detail here if the reviewer would like additional clarification.
>
> > *[...] include more diverse RLVR benchmarks other than the math benchmarks.*
>
> We are currently running experiments on a challenging code generation benchmark and are happy to add the experiments and results to the revision. Preliminary results suggest that TROLL provides similar benefits on code generation as on math-related tasks.
>
> > *Why the authors claim that TROLL is for LLMs only?*
>
> We don’t. TROLL consists of a discrete version of differentiable trust region projections [5] combined with a sparsification scheme for high-dimensional action spaces, both of which are highly relevant for LLMs. We agree that other applications, such as VLA training or even more classical discrete RL are interesting applications of TROLL. We will clarify TROLL’s scope in the revision.
>
> > *[...] it seems that the dual step size needs to be calculated for every token?*
>
> It does. In general, a per-token treatment seems to be crucial for LLM post-training due to the high importance of a few select tokens. Providing an efficient, batched dual optimization and closed-form gradients through this optimization are core contributions of our work.
>
> > *Why the gain for GSPO much larger than GRPO and DrGRPO?*
>
> As briefly mentioned in the caption of Table 1, we found GSPO training with clipping to be unstable in our experiments. Figures 7 (Qwen3-8B) and 8 (Qwen2.5-7B) in the appendix plot success rates during training. Both show that GSPO with clipping starts off well, but diverges in later training stages. We will clarify this connection in the revision.
>
>
> We want to thank the reviewer again for their important and detailed feedback. We encourage them to reach out during the rebuttal if any concerns remain unaddressed.
>
> [1] Huang, Shengyi, et al. "The 37 implementation details of proximal policy optimization." *ICLR Blog Track* (2023)
>
> [2] Engstrom et al., “Implementation Matters in Deep Policy Gradients: A Case Study on PPO and TRPO”, *ICLR* (2020)
>
> [3] Song, H. Francis, et al. "V-mpo: On-policy maximum a posteriori policy optimization for discrete and continuous control." *arXiv* (2019)
>
> [4] Xi, Zhiheng, et al. "BAPO: Stabilizing Off-Policy Reinforcement Learning for LLMs via Balanced Policy Optimization with Adaptive Clipping." *arXiv* (2025)
>
> [5] Otto, Fabian, et al. "Differentiable Trust Region Layers for Deep Reinforcement Learning." *ICLR* (2021)

---

### Official Review · Reviewer_uGtd · 2025-11-04

**Soundness:** 4
**Presentation:** 4
**Contribution:** 3
**Rating:** 10
**Confidence:** 4

**Summary:**

This paper introduces a novel method that replaces the standard PPO-like clip objective used in RL-based LLM fine-tuning. The authors posit that PPO's clipping is a crude approximation of a more principled KL-based trust region. TROLL substitutes this heuristic with a discrete differentiable trust region projection. For each token, the method solves a convex optimization problem that projects the new policy's output distribution onto a KL-ball centered around the old (sampling) policy's distribution. To make this practically feasible for large vocabularies, the authors additionally sparsify the predictive distribution and operate only on a small subset of the most probable logits, capturing the majority of the probability mass. Using implicit differentiation, this projection is fully differentiable, allowing gradients to propagate, unlike the gradient-cutting nature of PPO's clip. Experiments on mathematical reasoning benchmarks (DAPO-Math, GSM8K) show that TROLL consistently outperforms clipping in training stability, speed, and final success rates across various models (e.g., Qwen3, LLaMA 3).

**Strengths:**

Novel and innovative approach to trust region based RL under the practical constraints of modern LLM finetuning.
The approach is theoretically well motivated and the paper shows convincing empirical results.
The paper is very well written; the appendix with detailed deviations is easy to follow.
Experiments and empirical evidence is collected using a suite of differently open-weight model sizes and families; generally with very consistent and impressive results.

**Weaknesses:**

The experimental section focuses on consistently outperforming established benchmarks using a wide range of models and multiple datasets. This unfortunately leaves little space for more detailed analysis and ablations. The appendix D.6 however has at least some interesting additional detailed analysis.

**Questions:**

I am surprised about the low 0.1% clipping/projection rate displayed in Figure 14. Have you run ablations with different batch-sizes and model-staleness to better understand the effects of being on- vs. slightly off-policy?

Have you investigated how the policy entropy evolves over the course of learning compared to baseline methods?

---

> ### Comment · Reviewer_uGtd · 2025-11-12
> **Mistake**
>
> Dear authors, Dear SAC,
>
>
> When writing and entering the review I made a mistake and accidentally submitted a overall "10" recommendation instead "8".
>
> I do believe it is a good paper, and I'm looking forward to the rebuttal phase to clarify some details. But my intended starting point for the rebuttal was "8".
>
> I updated my rating accordingly.
>
> Apologies for the confusion.

---

> ### Author Response · Authors · 2025-11-20
> **Response to Reviewer uGtd**
>
> We thank the reviewer for their insightful feedback, and in particular for acknowledging the novelty and potential impact of our approach. In the following, we address the individual concerns raised.
>
> > *[The experiment section’s focus on comparing TROLL to CLIP] unfortunately leaves little space for more detailed analysis and ablations.*
>
> We fully agree with the reviewer that the current experiment section focuses heavily on performance, and that a stronger analysis would help explain TROLL’s behavior and provide valuable insights into the method. We are currently running evaluations on different batch sizes to evaluate how TROLL deals with more gradient steps per iteration. We are happy to add these results to the existing analysis in Section 5.3 and Figure 5 in the main paper, and to potentially surface some of the results of Appendix D.6. to the main discussion.
>
>
> > *I am surprised about the low 0.1% clipping/projection rate displayed in Figure 14. Have you run ablations with different batch-sizes and model-staleness to better understand the effects of being on- vs. slightly off-policy?*
>
> The low clipping/projection is an interesting point. We hypothesize that this low value is explained by a relatively small number of critical tokens during RL finetuning. For example, the model is unlikely to change the general layout of an answer, its grammar, or notation. Tokens that are responsible for these aspects do not seem to change enough to hit the clipping or projection bounds. The GSPO paper [1] similarly finds that GRPO’s token-level clipping only affects a very small percentage of tokens.
>
> > *Have you investigated how the policy entropy evolves over the course of learning compared to baseline methods?*
>
> The right side of Figure 5 in the main paper shows an entropy comparison between TROLL and CLIP for GRPO using Qwen3-14B. We find that TROLL’s entropy stabilizes over the course of the training, while the entropy keeps decreasing for CLIP. We agree that the entropy provides important insights into the learning dynamics, and we will highlight these results in the revision.
>
> We want to thank the reviewer again for their insightful feedback and positive reception, and look forward to further discussion during the rebuttal.
>
> [1] Zheng, Chujie, et al. "Group sequence policy optimization." arXiv (2025).

---

### Author Response · Authors · 2025-11-20
**General Response**

We thank all reviewers for the insightful comments, critical assessment, and helpful feedback on our work. In this response, we aim to pool common concerns and give a high-level overview of points raised by multiple reviewers, as well as additional experimental results and important changes to the manuscript during the revision.

**Additional experiments**:
* Multiple reviews (**cNjy**, **YCtx**) mention that TROLL’s current evaluation on math-related reasoning tasks should be expanded to a different application. We are currently running experiments on a code generation benchmark. Preliminary results suggest that TROLL provides similar improvements over clip as in the math tasks, and we look forward to sharing the full experimental evaluation later in this rebuttal period.
* Several reviewers (**cNjy**, **RHmH**) directly or indirectly suggest additional baselines that clarify the relation between TROLL, other methods that improve clipping, and clipping-free approaches. These experiments are also still running, but so far seem to show that TROLL’s projection provides a unique advantage over other clipping variants and clipping-free alternatives. We hypothesize that this advantage stems from the inherent differentiability of TROLL’s projection, as well as its relatively accurate implementation of the underlying trust region constraint.
* We started experiments with larger batch sizes to evaluate how TROLL behaves and compares to clipping when using more update steps per iteration (**uGtd**, **RHmH**). So far, it seems that TROLL stays stable for higher batch sizes, while clipping slowly degrades.

We will provide another update once these results are fully available, and will add detailed experimental setups and results to the revised paper and its appendix.

**Presentation**:
* Reviewers **cNjy** and **RHmH** voiced concerns about the clarity of our presentation. We thank them for their feedback and suggestions for improvements. We have added relevant related work, and revised the method section to include pseudocode and a schematic on gradient flow. We also clarified the relationship between the different objectives, and made minor improvements to the notation to improve clarity.
* Two reviewers (**cNjy**, **YCtx**) state that the performance benefits of TROLL are “marginal”, while the other reviewers think they are “very consistent and impressive” (**uGtd**) and “showcase the advantage of the proposed approach in many cases” (**RHmH**). The "marginal" perception likely arises from comparing models of different size or looking at absolute gains. Aggregating over our experimental results, we find that TROLL generally provides a substantial 3-10% absolute improvement in most settings, which corresponds to 5-15% relative, depending on the setup. Here, using TROLL over clipping is more advantageous than tuning the advantage estimation algorithm. These results are further supported by the additional experiments mentioned above. In some cases, such as GSPO training in Table 1 or Llama3.1-8B/Apertus-8B in Figure 4, TROLL provides a clear and well-behaved optimization, while the clip objective completely fails. In all these cases, we argue that TROLL’s improvements are significant, especially since it simply replaces clipping. Further, we evaluate TROLL both in terms of update steps (e.g., Figure 3) and wall clock time (e.g., Figure 1), showing that it outperforms clipping in both settings.

**Code**:
* While not explicitly requested, we are happy to provide the codebase used for TROLL’s experiments in the supplement and to fully open-source it after de-anonymization. We believe that this codebase will allow researchers and practitioners to use TROLL as a direct replacement for clipping-based approaches.

Finally, we encourage all reviewers to reach out to us if any questions or concerns remain unaddressed, and we look forward to providing the revised version of the manuscript once the last experiments have finished.

---

### Author Response · Authors · 2025-11-23
**Upload of First Revision**

Dear reviewers,

We are happy to provide a first revision of our paper. We clarified the method section and included pseudocode on TROLL's policy update in the main paper, as well as an overview of gradient flow in the appendix. We also revised the related work, adding discussion on several methods that utilize adaptive clipping, or omit computing importance ratios and thus clipping altogether. Finally, our presentation on TROLL's experimental performance now highlights its improvement over clipping in several scenarios. We marked these sections in orange and also made minor updates in other places to ensure consistency.

We want to thank the reviewers again for their valuable feedback. We will upload the experimental results in the following week, and look forward to further discussion during the rebuttal period.

---

### Author Response · Authors · 2025-11-27
**Upload of Second Revision (Additional Experiments)**

We are happy to provide another revision of our paper, presenting additional experimental results:

* Code Generation: We present results on code generation tasks using the Eurus-2-RL-Code benchmark, Qwen 3 models of different sizes, and GRPO. Similar to the existing results, these experiments show how TROLL significantly improves performance, in this case by 7-18 percentage points, which translates to more than a 20% relative gain for each model size.

* Comparison to a clipping-free approach: We compare against an algorithm that does not require clipping, namely GPG. While GPG diverged in our setting, using TROLL resolves the divergence issues, achieving parity with GRPO. This initial result shows how TROLL can be applied beyond clipping-based objectives, as it operates directly on the LLM output and not the ratio.

* Comparison to an approach with adaptive KL constraints: We compare against the very recent BAPO approach, and show that it does not reach TROLL's performance.

* Additional Analysis: In addition to the existing entropy analysis, we added further entropy results using the newly added code generation benchmark. Additionally, we include an analysis of the batch size as a proxy for the number of updates between generating new sequences, showing that TROLL is robust to this hyperparameter while clipping degrades with larger batch sizes.

* We also included Reinforce++ as another GRPO-adjacent method to further evaluate TROLL's general usefulness as a replacement to clipping.

Finally, we restructured some of the existing experiments' presentation to further highlight TROLL's benefits and our analysis of different design choices and metrics beyond reward.

We want to thank the reviewers again for their valuable suggestions regarding additional experiments, and strongly believe that these experiments help to highlight TROLL’s efficacy. We hope that the new results address the concerns of the reviewers with respect to the experimental scope and significance, and look forward to further discussions.

**EDIT December 01:** Slightly updated Code Generation result after convergence of all models.

---

### Author Response · Authors · 2025-12-01
**Response to Changes in Review Process**

Dear Area Chair and other members of the committee

Due to recent issues with OpenReview and the resulting change in the reviewing process, we wanted to outline the course of our rebuttal period briefly. The initial reviews for our paper were overall pretty positive, finding three main weaknesses with the manuscript
- Reviewers **cNjy** and **RHmH** mentioned missing related work and the potential for additional baselines.
- The reviewers were concerned about marginal performance gains (**cNjy**, **YCtx**), limited analysis (**RHmH**, **cNjy**, **uGtd**), and math-only experiments(**cNjy**, **YCtx**).
- Reviewers **cNjy** and **RHmH** voiced concerns about the clarity of our presentation, particularly the method section and the optimization procedure of TROLL.

We addressed these concerns in our responses to the individual reviewers and revised the paper twice, incorporating the reviewers’ feedback. We marked changes in the manuscript in orange. We also added our code to the supplement for reproducibility.
For the first revision, uploaded at the beginning of last week, we revised related work and the description of our method. In particular, we added discussions on alternatives to clipping and extended our related work section. We also resturcuted the method section, clarified notation, and provided pseudocode. These changes were positively acknowledged by two of the reviewers (**YCtx**, **RHmH**) while the others did not comment before this was disabled.

In the second revision, uploaded at the end of last week, we added novel experimental results.
- We added evaluations on a code generation task using models from the Qwen 3 family. On these tasks, TROLL performs especially well, improving over standard clipping by 7-18 percentage points, depending on model size. These experiments show TROLL’s applicability beyond mathematical reasoning.
- We include three additional baselines, namely REINFOCE++ as another advantage estimation method, BAPO as a recent adaptive clipping method suggested by **cNjy**, and GPG as a clipping-free method akin to those proposed by **RHmH**.  For all these experiments, we found that TROLL improves over baselines in terms of success rate and training stability.
- We include an analysis of TROLL’s behaviour for different batch sizes, additional entropy evaluations for the new code generation results, and slightly restructured the manuscript to highlight our analysis of TROLL beyond pure performance gains (**uGtd**).

While the reviewers had unfortunately no opportunity to comment on these changes due to the changes in the review process, these new results clearly show TROLL’s efficacy and usefulness to the field of Reinforcement Learning from Verifiable Feedback tasks.

Overall, we believe incorporating the reviewers' feedback and addressing their concerns further improved our already well-received submission. We would like to thank the reviewers again for their feedback and kindly ask that our revisions be taken into account when making a decision.

---

### Meta-Review · Area_Chair_SzvN · 2026-01-06

**Summary:**

This paper proposes replacing the, somewhat unprincipled, clipping that PPO uses with a differentiable trust region projection. The proposed method is theoretically well-founded and sound, and the authors provide a reasonable approximation which makes the approach tenable in practice. The empirical results provide strong evidence for the validity of the proposal, and all reviewers are generally positive about the work.

The reviewers raised a number of concerns during the review process, but the authors did a remarkable job addressing all of them. These included:
- Running extra evaluations on a non-math problem (code generation), including extra entropy analyses
- Addition of an extra baseline (GPG)
- Addition of a very recent approach (BAPO). This is commendable, as this approach was put on arxiv after the submission deadline.
- Additional evaluation of TROLL on Reinforce++

Overall, I feel like this submission is a valuable contribution to the community. It was likely an accept prior to the rebuttal, but the submission is substantially stronger after the author's incorporated all of the reviewer feedback.

As such, I give a strong recommendation of acceptance of this work.

**Reviewer Concerns:**

I believe all reviewer concerns were addressed. The ones which stand out as most significant are:
- Adding non-math evaluation (cNjy, YCtx)
- Adding more analyses beyond performance (uGtd)
- Clarifying section3, such as by adding pseudocode (cNjy)
- Adding extra baselines (cNjy, RHmH). It is worth mentioning that the authors would have been justified in *not* running the BAPO baseline, as BAPO was uploaded to arxiv after the submission deadline; thus, it is commendable that the authors were able to include this baseline.
- Providing more ablations and analyses to better understand TROLL's performance (uGtd, RHmH)
- Improved presentation (all reviewers)

**Reviewer Scores:**

- **uGtd**: currently has an 8, so would have likely maintained this score
- **cNjy**: currently has a 4, but all concerns were adressed, so they would have likely increased to a 6
- **YCtx**: currently has a 6, although they specified that they were increasing their score, so it's not clear if 6 is the score after increase or not. in any case, all concerns were addressed, so their final score would have been a 6 or an 8.
- **RHmH**: currently has a 6. all concerns were addressed, so would have likely stayed at 6 or increased to 8.

---

### Decision · Program_Chairs · 2026-01-26

Accept (Oral)